# Regulatory Genetic Networks by microRNAs: Exploring Genomic Signatures in Cervical Cancer

**DOI:** 10.3390/biomedicines13061457

**Published:** 2025-06-13

**Authors:** Carlos Pérez-Plasencia, Yaneth Citlalli Orbe-Orihuela, Armando Méndez-Herrera, Jessica Deas, Claudia Gómez-Cerón, Hilda Jiménez-Wences, Julio Ortiz-Ortiz, Gloria Fernández-Tilapa, Aldo Francisco Clemente-Soto, Jesús Ricardo Parra-Unda, Jesús Salvador Velarde-Felix, Mauricio Rodríguez-Dorantes, Oscar Peralta-Zaragoza

**Affiliations:** 1Unidad de Biomedicina, FES-Iztacala, Universidad Nacional Autónoma de Mexico, Tlalnepantla de Baz 54090, Mexico; carlos.pplas@gmail.com; 2Dirección de Infecciones Crónicas y Cáncer, Centro de Investigación Sobre Enfermedades Infecciosas, Instituto Nacional de Salud Pública, Cuernavaca 62100, Mexico; yaneth.orbe@insp.mx (Y.C.O.-O.); amendez@insp.mx (A.M.-H.); jessicadeas@gmail.com (J.D.); 3Departamento de Epidemiología del Cáncer, Centro de Investigación en Salud Poblacional, Instituto Nacional de Salud Pública, Cuernavaca 62100, Mexico; cceron@insp.mx; 4Laboratorio de Investigación Clínica, Unidad Académica de Ciencias Químico-Biológicas, Universidad Autónoma de Guerrero, Chilpancingo 39070, Mexico; wences2009@hotmail.com (H.J.-W.); julioortiz@uagro.mx (J.O.-O.); gferti@hotmail.com (G.F.-T.); 5Facultad de Ciencias Químico-Biológicas, Universidad Autónoma de Sinaloa, Culiacán Rosales 80040, Mexico; aldoclemente@uas.edu.mx (A.F.C.-S.); ricardoparraund@uas.edu.mx (J.R.P.-U.); 6Facultad de Biología, Universidad Autónoma de Sinaloa, Culiacán Rosales 80040, Mexico; jsvelfe@hotmail.com; 7Laboratorio de Oncogenómica, Instituto Nacional de Medicina Genómica, Tlalpan 14610, Mexico; mrodriguez@inmegen.gob.mx

**Keywords:** regulatory genetic networks by microRNAs in cervical cancer

## Abstract

Cervical cancer remains a significant global health concern, impacting over half a million women annually. The primary cause is a persistent infection with hr-HPV, which disrupts various cellular processes crucial for normal function. This disruption leads to genetic instability, including changes in the expression of microRNAs and their corresponding host genes, with far-reaching consequences for cellular regulation. Researchers have widely utilized high-throughput technologies to analyze gene expression in cervical cancer, aiming to identify distinct molecular signatures of microRNAs and genes through genomic analysis. However, discrepancies among studies have been noted, possibly due to variations in sample collection, technological platforms, and data processing methods such as normalization and filtering. Therefore, it is essential to synthesize findings from diverse studies to comprehensively understand the molecular mechanisms of regulatory genetic networks involved in the initiation and progression of cervical cancer. This review examined the evidence detailing the role of microRNA signatures and their target genes in cervical carcinogenesis and disease advancement. The accumulated data suggest the presence of widespread regulatory genetic networks active in both precancerous and cancerous cervical cells, potentially acting as key drivers of this malignancy. Identifying these molecular genomic signatures could open new avenues for developing therapeutic strategies for cervical cancer, particularly in the realm of precision medicine.

## 1. Introduction

Cervical cancer is a significant global health concern, ranking as the fourth most common malignancy among females and the seventh overall. Developing nations bear a disproportionately high burden, accounting for over 85% of cases. This cancer comprises 13% of all female cancers, and when combined with other HPV-associated malignancies, it represents 5.2% of all human tumors [1]. The primary cause of cervical cancer is human papillomavirus (HPV) infection. While most infections are cleared by the immune system, some persist and can lead to the progression of cervical intraepithelial neoplasia (CIN) through various stages (CIN I, II, and III). Following CIN III, malignant cells can invade the pelvic cavity, leading to cervical cancer [2]. Current primary screening for cervical cancer predominantly relies on cervical cytology. However, this method has limitations, including technical challenges, variability between observers, and inherent issues with the pathomorphological classification system. These factors contribute to its relatively low and inconsistent sensitivity (50–80%) and compromised specificity [3]. Therefore, there is a need for objective, high-throughput, and feasible triage tests, especially for resource-limited settings. Various screening approaches have been adopted, combining cytology, HPV testing, oncogenic HPV genotyping, and genomic signatures [4]. While persistent high-risk oncogenic HPV (hr-HPV) infection is a critical initiating factor, additional elements influence the development and progression of cervical cancer. Genetic variations at susceptibility loci and genetic instability have been observed [5]. Perturbations in cellular gene expression are also linked to HPV-associated carcinogenesis [6]. Despite advances in screening and treatment, some patients still experience metastatic disease or resistance to radiotherapy and chemotherapy [7,8]. Consequently, a deeper understanding of the molecular mechanisms driving cervical cancer is vital for developing new diagnostics and personalized treatments.

The complex regulatory genetic networks involved in the progression from HPV infection to tumor development are not yet fully understood. Genomic signatures, such as altered microRNA expression profiles, are crucial areas of investigation as they are implicated in this complex disease. MicroRNAs are non-coding RNAs that regulate gene expression by binding to mRNAs, leading to their cleavage or translation repression. In this mechanism, microRNAs modulate gene expression at the post-transcriptional level by binding to 3′-untranslated regions (3′-UTRs) of mRNAs, leading to either target mRNA cleavage or repression of mRNA translation [9]. MicroRNA-mediated regulatory genetic networks may offer prognostic utility and predictive capabilities for patient survival, potentially contributing to precision medicine [10].

Several previous studies have identified microRNAs as potential targets for different therapeutic strategies associated with HPV infection or have linked specific microRNAs to the cervical carcinogenesis process. By perming Medline, we search for the keywords cervical cancer and microRNA signature (14 April 2021). In this work, only the studies that indicate the platforms used for microRNA analysis, several experimental conditions, regulatory genetic networks modulated by microRNAs, microRNA-mRNA regulatory modules, and clinical trials with microRNAs for uterine cancers were included. In this review, 70 different studies reported with different microRNA signatures, 15 databases, and nine clinical trials were included in this study. The microRNA signatures experimental reports were analyzed, which were proposed by several research groups to be associated with cervical cancer. Figure 1 shows a detailed description of the methodology used to select the studies included in this review article. We analyzed from these publications the microRNAs that were reported to be differentially regulated in cervical cancer tissue or in vitro models. Investigating the cellular microRNA composition and regulatory genetic networks in HPV-positive cancer cells in relation to endogenous E6 and E7 expression levels is crucial, as sustained E6 and E7 expression leads to relevant cellular phenotypes.

Despite significant findings on microRNA signatures and cervical cancer, a comprehensive understanding of the central molecular mechanisms requires an integrative multi-omics approach. The computational integration of biomolecular networks with data from various omic levels is central to research in regulatory genetic networks, offering insights into genome reprogramming and potential diagnostic or therapeutic targets. Investigating target hubs regulated by multiple microRNAs is important as they often act as key regulators of biological processes. MicroRNA-mediated gene regulatory networks are increasingly recognized as critical for understanding cooperative microRNA regulation in cancer, with their complexity arising from numerous microRNA-target interactions and feedback loops.

## 2. Genomic Signatures of microRNAs in Cervical Cancer

Non-coding RNA (ncRNA) denotes RNA species that do not undergo translation into protein. NcRNA can be broadly classified into two main groups: housekeeping ncRNA and regulatory ncRNA, the latter encompassing short-chain siRNA, microRNA, piwiRNA, and lncRNA. MicroRNAs constitute a large class of small (19–24 nucleotides), highly conserved, single-stranded ncRNAs present across a wide range of eukaryotes and viruses. A notable 50% of microRNAs are localized to chromosomal regions exhibiting a propensity for structural alterations. MicroRNAs exert post-transcriptional regulatory control over mRNA targets through both classical and noncanonical mechanisms [11]. Specifically, microRNAs bind to the 3′-UTRs of target genes via the seed sequence, leading to either direct mRNA cleavage (in cases of full complementarity) or translational repression (in cases of incomplete complementarity) [11]. Previous investigations have established the involvement of microRNAs in diverse biological processes, including cell growth, development, differentiation, apoptosis, and cellular homeostasis. Furthermore, microRNA dysregulation has been implicated in tumorigenesis and tumor progression, notably by fostering uncontrolled proliferation, enhancing survival, impeding differentiation, and promoting cancer cell metastasis, as observed in cervical cancer [12]. These findings support the proposition that systems biology-based cooperating microRNAs may represent a viable gene therapy approach for cancer.

Bioinformatic analysis of sequences using various online platforms, including miRTarBase v2025 [13], TargetScan v7.2 [14], miRanda v3.3a [15], miRbase v2925 [16], MirWalk v3 [17], TriplexRNA v2023 [18], miRSearch v2025 [19], Eclipsebio v2024 [20], miR2disease v2024 [21], DIANA-miRPath v4.0 [22], GeneCodis v4 [23], miRecords v4 [24], and TransmiR v2.0 [25], indicates that a single microRNA may possess binding sites on hundreds of mRNAs, and conversely, a single mRNA 3′-UTR may be targeted by tens of microRNAs. Post-transcriptional regulatory element sequencing facilitates the identification of microRNA and RNA-binding protein target sequences within mRNAs [26]. Dysregulated microRNA expression, when compared to normal cells, constitutes a key mechanism influencing the roles of microRNAs in cancer development. The direct regulatory action of microRNAs on target mRNA is subject to modulation by factors beyond the microRNA itself. It has been observed that a single microRNA sequence may regulate distinct targets at different stages of carcinogenesis, a phenomenon designated as “microRNA’s differential regulation”. Furthermore, the precise definition of effective target interactions and related process outcomes is contingent upon the concordance of alterations in microRNA regulation and mRNA contribution. The capacity of microRNAs to regulate diverse targets across different stages may represent a widespread phenomenon. Consequently, microRNAs are essential components of genetic regulatory networks, functioning in the post-transcriptional regulation of cellular gene expression.

### 2.1. Genomic Signature of microRNAs in Cervical Tissues

MicroRNAs are emerging as promising biomarkers for cancer classification and predicting outcomes across various human cancers. The initial microRNA expression signature in cervical cancer was identified by Hu et al. [27]. This research pinpointed a microRNA-based signature for predicting cervical cancer survival. The study involved analyzing the expression profiles of 96 cancer-related microRNAs in 102 cervical cancer samples, utilizing RT-qPCR-based microRNA assays. The findings revealed that five specific microRNAs (miR-9, miR-21, miR-200a, miR-203, and miR-218) were significantly linked to patient survival. Among these, miR-9 and miR-200a showed particular potential as candidates for a prognostic microRNA signature, demonstrating strong associations with disease outcomes. Further functional studies were conducted to understand the impact of these microRNAs on cervical cancer cells. Through integrative analyses, including genome-wide microRNA target prediction, real-time RT-qPCR, and cell migration assays, evidence suggested that miR-200a affects the metastatic potential of cervical cancer cells. This occurs by simultaneously suppressing multiple genes involved in cell morphology and migration sites. These results imply that miR-9, miR-21, miR-200a, miR-203, and miR-218 could form a microRNA signature, offering a novel biomarker-driven approach for risk stratification in cervical cancer patients.

A study by Pereira et al. [28] explored microRNA expression patterns in diverse cervical tissues from 25 patients. They employed a microarray platform to analyze microRNA expression signatures in cervical cancer, using probes for mature microRNAs. This research identified eight microRNAs that were significantly downregulated during the progression from normal cervical samples to pre-neoplastic and neoplastic states, specifically miR-26a, miR-29a, miR-99a, miR-143, miR-145, miR-199a, miR-203, and miR-513. Conversely, other microRNAs showed downregulation from normal to pre-neoplastic cervical samples but then exhibited increased expression in cervical cancer samples. These included miR-16, miR-27a, miR-106a, miR-142-5p, miR-197, and miR-205. Additionally, miR-10a, miR-132, miR-148a, miR-196a, and miR-302b were found to be upregulated in CIN I, CIN III, and cervical carcinoma samples. The natural variability in microRNA expression profiles observed could be partly attributed to biological heterogeneity among cervical samples, which might complicate the use of microRNA profiling in clinical diagnostics.

Recent research has shed light on the involvement of microRNAs in the progression of CIN to cervical carcinoma. A study identified a unique microRNA expression pattern associated with CIN, pointing to specific microRNAs that may play a role in the development of cervical cancer [29]. In this research, high-grade CIN II/III tissues were meticulously isolated and contrasted with healthy cervical epithelial tissue. To ensure the precise and sensitive profiling of 202 microRNAs, RT-qPCR was employed. A focused analysis revealed that 12 particular microRNAs were significantly dysregulated, effectively distinguishing high-grade CIN samples from normal cervical epithelium. These microRNAs included miR-9, miR-10a, miR-20b, miR-34b, miR-34c, miR-193b, miR-203, miR-338, miR-345, miR-424, miR-512-5p, and miR-518a. Further predictive analysis suggested that these dysregulated microRNAs primarily influence apoptosis signaling pathways and cell cycle regulation. These findings validate the connection between this microRNA signature and an independent cohort of high-grade CIN cases, also demonstrating its effectiveness in differentiating cervical squamous cell carcinoma from healthy controls.

Analysis of microRNA expression profiles has revealed distinct signatures associated with cervical cancer and its precursor, CIN [30]. Researchers utilized microarrays to compare microRNA expression in normal cervical tissues, CIN, and cervical cancer, with subsequent validation via RT-qPCR. In cervical squamous cell carcinoma, compared with normal tissue, miR-15b, miR-16, miR-21, and miR-21* were found to be significantly overexpressed. Conversely, miR-218 and miR-376 were the most notably downregulated miRNAs in these tissues. For CIN III tissues, an upregulation of miR-188-5p, miR-483-5p, miR-663, miR-765, and miR-1300 was observed when compared to normal cervical tissues. In contrast, miR-142-3, miR-149, miR-152, miR-218, miR-374b, and miR-376c were significantly downregulated in CIN III. Further validation studies indicated consistent upregulation of miR-9, miR-21, and miR-31 across both CIN III and cervical cancer. Simultaneously, miR-195, miR-199b-5p, miR-218, miR-376a, and miR-497 were consistently downregulated in both conditions. MiR-218 emerged as the most consistently downregulated miRNA, suggesting a potential role as a tumor suppressor in cervical cancer by targeting various cancer-related genes. In contrast, miR-21 was the most consistently upregulated miRNA, likely functioning as an oncogene.

Investigating microRNA expression profiles in cancer holds promise for guiding treatment decisions, particularly in cervical cancer. However, the direct clinical application of these findings faces significant hurdles. One major challenge lies in the variability of sample types. While some studies effectively analyze miRNA expression in formalin-fixed paraffin-embedded (FFPE) specimens, inconsistencies can arise when comparing data from frozen and FFPE samples [31]. This non-comparability can undermine efforts to validate prognostic signatures. Another critical issue is the inherent heterogeneity within tumors. The genetic and molecular differences across various regions of a single tumor can lead to discrepancies in microRNA expression, making it difficult to establish a universally applicable signature. Furthermore, the current technological platforms used for global microRNA expression profiling may lack the necessary robustness for clinical application. This limitation could potentially be mitigated by employing advanced techniques like deep sequencing for more comprehensive and accurate microRNA expression analysis. The unsuccessful validation of a specific 9-microRNA prognostic signature for cervical cancer, with only miR-1 and miR-21 upregulated, despite its initial identification in frozen specimens, underscores these challenges.

Understanding how the E6 and E7 oncogenes of HPV influence the microRNA landscape in cervical cancer cells is a key area of research. Honegger et al. investigated the impact of E6a and E7 expression on both intracellular and exosomal microRNA profiles in HPV-positive cancer cells [32]. Their deep-sequencing analysis aimed to characterize the global microRNA composition, with validation of candidates performed using RT-qPCR. Silencing E6 and E7 in HeLa cells (HPV18+) led to notable changes in intracellular microRNAs. Ten of the 52 most abundant intracellular microRNAs were significantly affected: miR-7-5p, miR-17-5p, miR-186-5p, miR-378a-3p, miR-378f, and miR-629-5p were downregulated, while miR-23a-3p, miR-23b-3p, miR-27b-3p, and miR-143-3p were upregulated. These E6 and E7-regulated microRNAs are associated with cellular processes like proliferation, senescence, and apoptosis, suggesting their role in the growth of HPV-positive cancer cells. In exosomes secreted by HeLa cells, a distinct signature of seven microRNAs was identified among the most abundant. Upon E6 and E7 silencing, let-7d-5p, miR-7-5p, miR-20a-5p, miR-92a-3p, miR-378a-3p, and miR-423-3p were significantly downregulated, while miR-21-5p was upregulated. Several of these E6 and E7-dependent exosomal microRNAs are also known to be involved in controlling cell proliferation and apoptosis. Interestingly, the microRNA percentage in exosomes was about 50% lower compared with intracellular small RNA fractions, indicating compositional differences. These findings indicate that HPV E6 and E7 oncogene expression shapes a specific seven-microRNA signature in exosomes from HeLa cells, leading to increased levels of let-7d-5p, miR-7-5p, miR-20a-5p, miR-92a-3p, miR-378a-3p, and miR-423-3p, and decreased miR-21-5p.

A new microRNA has been identified and linked to cervical cancer [33]. Researchers investigated the role of miR-466, specifically its functions and targets, within cervical cancer tissues. The study included 157 participants, categorized into three groups: 56 cervical cancer patients, 60 patients with CIN, and 49 healthy individuals. Higher levels of miR-466 expression were noted in both cervical cancer and CIN patients compared with healthy controls. Furthermore, miR-466 expression was elevated in patients who had lymph node involvement. However, no significant correlation was found between miR-466 expression and FIGO stages, tumor size, or vascular invasion. These findings suggest a connection between abnormal miR-466 expression and the development and progression of cervical cancer, though it may not serve as a marker for tumor characteristics.

Researchers have investigated the molecular signatures of microRNAs in cervical precursor lesions, aiming to identify pathways associated with cervical cancer progression [34]. The study involved categorizing samples into those with CIN III or those without precursor lesions. MicroRNA and gene expression profiling utilized nCounter miRNA Expression Assays and Pan Cancer Pathways, respectively. To understand the role of these microRNAs, target prediction was performed using mirDIP, and molecular pathway interactions were mapped with Cytoscape v3.10.3. The study employed bidirectional in silico analyses, and Pearson’s correlation to determine associations between genes’ differentially expressed microRNAs and the progression of cervical cancer. Nine microRNAs were identified as potential biomarkers, with two (miR-381-3p and miR-4531) being significantly downregulated and seven (let-7f-5p, miR-128-2-5p, miR-130a-3p, miR-202-3p, miR-205-5p, miR-323a-5p, and miR-3136-3p) significantly upregulated in CIN III. Interestingly, these microRNA expression patterns were found to be independent of hr-HPV infection. Specifically, four microRNAs (miR-130a-3p, miR-205-5p, miR-381-3p, and miR-4531) were highlighted as potential biomarkers for CIN III in liquid-based cytology samples. Correlations were observed between certain microRNAs and gene expression: miR-130a-3p showed a negative correlation with CCND1, miR-205-5p a positive correlation with EGFR, and miR-4531 a negative correlation with SMAD2. This panel of microRNAs may play a role in regulating key molecular pathways involved in carcinogenesis and could serve as a biomarker to distinguish CIN III from healthy cervical tissue in liquid-based cytology samples.

MicroRNAs have been investigated for their role in cervical cancer prognosis. Researchers utilized publicly available data from TCGA, specifically microRNA expression profiles and clinical data from cervical cancer patients [35]. An integrated bioinformatics strategy was employed to pinpoint prognostic microRNAs. The microRNAs chosen for further validation were those demonstrating the most significant synergistic and additive effects. Notably, three specific microRNAs (miR-216b-5p, miR-585-5p, and miR-7641) were found to be highly effective in predicting a poor prognosis through additive effects analysis. Beyond identifying these markers, functional enrichment analysis revealed that both conventional cancer pathways and immune system pathways likely play a significant role in determining disease outcomes. This suggests that these microRNAs may exert their influence through mechanisms involving these critical biological pathways.

Identifying robust biomarkers for cancer prognosis is crucial for improving patient outcomes. Research has highlighted the potential of microRNA signatures in various cancers, including cervical cancer. A study investigated microRNA expression patterns in cervical cancer by analyzing high-throughput data from the TCGA database, comparing cancerous tissues to normal cervical tissues [36]. This analysis led to the identification of 78 differentially expressed microRNAs. Among these, 37 were found to be upregulated, and 41 were downregulated in cervical cancer tissues. Further evaluation focused on the prognostic significance of these differentially expressed microRNAs, ultimately leading to the development of a three-microRNA signature. This signature, comprising miRNA-145, miRNA-200c, and miRNA-218-1, demonstrated a strong ability to predict patient survival in cervical cancer.

Recent research has illuminated the role of microRNAs in cervical cancer development. Scientists identified a unique microRNA expression pattern in cervical cancer, suggesting their involvement in the disease’s progression [37]. This investigation involved profiling microRNA expression in 24 cervical tissue samples comprising 18 cancerous and six normal tissues. Advanced techniques such as next-generation sequencing and microRNA microarrays were employed to analyze microRNA profiles in both cervical cancer cell lines and tissue specimens. A specific microRNA signature was subsequently validated using RT-qPCR, and its biological implications were explored through various computational analyses. Thirteen distinct microRNAs were identified as a signature for cervical cancer. Among these, miR-21-5p, miR-135b-5p, miR-363-3p, and miR-429 showed overexpression in cervical cancer. Conversely, several microRNAs, including miR-136-5p, miR-218-5p, miR-377-3p, miR-497-5p, miR-1184, miR-3196, miR-4687-3p, miR-5587-3p, and miR-5572, were found to be downregulated. Computational analyses indicated that these microRNAs primarily influence crucial cellular pathways like PI3K-Akt and mTOR. The findings suggest that microRNAs associated with hr-HPV infections impact similar pathways regardless of the specific HPV type, implying a common carcinogenic mechanism across all hr-HPV infections. Functional analysis of the target genes for these three microRNAs indicated their involvement in several cancer-associated signaling pathways. These pathways include but are not limited to MAPK, AMPK, focal adhesion, cGMP-PKG, Wnt, and mTOR signaling pathways. The collective findings suggest that this specific three-microRNA signature could serve as a valuable prognostic marker in cervical cancer.

MicroRNAs are being explored for their utility in detecting cervical disease. One study identified a panel of six microRNAs that show promise as biomarkers for cervical cancer and pre-malignant lesions [38]. These microRNAs (miR-20a, miR-92a, miR-141, miR-183*, miR-210, and miR-944) were found to be significantly elevated in individuals with cervical abnormalities compared with healthy controls. This suggests an oncogenic role for these microRNAs in cervical carcinogenesis. The research indicates that these microRNAs could be valuable in distinguishing between healthy individuals and those with pre-malignant lesions or cervical cancer. Their increased expression in CIN also points to their potential involvement in the early stages of cervical cancer development. Among the six microRNAs studied, miR-141 demonstrated the highest sensitivity and specificity for identifying cancer patients. The overall microRNA signature offers significant diagnostic potential for early detection due to its high sensitivity and specificity.

A study investigated the potential of microRNAs as a triage tool in cervical cancer screening for women who tested positive for hr-HPV [39]. This research identified several microRNAs with altered expression levels in cervical scrapes. Specifically, three microRNAs showed increased expression (miR-9-5p, miR-15b-5p, and miR-28-5p), while five exhibited decreased expression (miR-100-5p, miR-125b-5p, miR-149-5p, miR-203a-3p, and miR-375). The study confirmed these expression changes using RT-qPCR, noting that the patterns observed in cervical tissue were consistent with those found in cervical scrapes. Five of these microRNAs showed significant differences in expression when comparing controls, CIN III, and cancer cases. To detect CIN III in hr-HPV-positive cervical scrapes from a screening population, a microRNA classifier was developed. This classifier, composed of miR-15b-5p and miR-375, demonstrated the ability to identify a substantial portion of CIN III cases and all carcinomas with 70% specificity. When combined with HPV16 and HPV18 genotyping, the rate of CIN III detection improved. The two-microRNA classifier achieved 55% sensitivity and 70% specificity for CIN III detection and 100% sensitivity for detecting cervical squamous cell carcinomas and adenocarcinomas. These findings suggest that using these differentially expressed microRNAs could offer an alternative molecular strategy for triage in hr-HPV-based cervical screening programs.

Ma et al. identified a microRNA expression signature capable of predicting survival time for patients with cervical squamous cell carcinoma [40]. Researchers identified a two-microRNA signature, consisting of miR-378c and miR-642a, that shows promise as a predictor of survival time in these patients. The investigation began with an analysis of 332 microRNAs in 261 cervical squamous cell carcinoma patients, divided into training and validation cohorts, utilizing data from the TCGA data portal. This led to the identification and subsequent validation of the microRNA expression signature. Further analysis involved constructing a regulatory network and performing gene target analysis, incorporating GO and KEGG. The regulatory network revealed 345 microRNA signature-target pairs, with 316 genes specifically targeted by miR-378c and miR-642a. A network of these 316 target genes was then constructed. Functional analysis of these target genes indicated significant enrichment in pathways such as MAPK, VEGF signaling pathways, and endocytosis. The findings suggest that this specific two-microRNA signature could serve as a valuable predictive factor for survival in patients with cervical squamous cell carcinoma.

A study investigated the role of specific microRNAs in cervical conditions, focusing on their potential as diagnostic and therapeutic tools [41]. Researchers analyzed microRNA expression in tissue samples from individuals with HSIL, SCC, and healthy controls. The microRNAs studied were initially identified as being associated with autophagy. Several microRNAs, including miR-30c, miR-143, miR-372, and miR-375, were found to be significantly downregulated in both HSIL and cervical SCC. Conversely, miR-130a showed a notable increase in expression, specifically within the cervical SCC group, when compared to both HSIL and control groups. Further analysis identified miR-30a, miR-520e, miR-548c, and miR-372 as important diagnostic indicators that also correlated with the overall survival rates of patients with cervical SCC. These findings collectively suggest that microRNAs linked to autophagy could play a significant role in both the diagnosis and targeted treatment strategies for cervical cancer.

In a study, researchers identified a distinct seven-microRNA signature that could be crucial for understanding and predicting the progression of cervical cancer [42]. This signature, which includes miR-144, miR-147b, miR-218-2, miR-425, miR-451, miR-483, and miR-486, was pinpointed through an extensive analysis of cervical cancer expression data from the TCGA database. The investigation focused on microRNAs that showed significant differential expression between early and advanced stages of the disease. An optimal subset of these microRNAs was then selected using a random forest algorithm to develop a cervical cancer-specific support vector machine (SVM) classifier. This SVM classifier, built upon the seven-microRNA signature, exhibited strong performance in predicting the pathological stages of patient samples. Beyond prediction, the study also delved into the functional implications of this microRNA signature within cervical cancer. The findings suggest that this specific set of microRNAs holds promise as a novel diagnostic and prognostic tool for cervical cancer.

Research has indicated that specific microRNAs, such as miR-34a, miR-193a, miR-200a, miR-423, and miR-455, form a distinct signature associated with cervical squamous cell carcinoma [43]. Studies have shown that miR-34a, miR-200a, and miR-455 are significantly uprexpressed in cervical squamous cell carcinoma tissues when compared to normal cervical tissue from healthy individuals. While miR-34a and miR-455 have demonstrated potential for diagnosing cervical cancer, miR-200a did not exhibit similar diagnostic significance. A particularly noteworthy finding is the strong correlation between low miR-34a expression and reduced overall survival rates in cervical cancer patients, establishing miR-34a as an independent prognostic indicator. These findings highlight the importance of these specific microRNAs as potential biomarkers for cervical cancer.

A systematic review of 27 studies involving 1721 cervical cancer cases and 1361 healthy control samples identified 26 microRNAs that exhibited differential expression across various stages of the disease [44]. The altered expression of these microRNAs was consistently verified through RT-qPCR. Specifically, 19 microRNAs were found to be downregulated in cervical cancer tissues compared with normal tissues. These include miR-1, miR-107, miR-132, miR-139-3p, miR-143, miR-195, miR-335-5p, miR-337-3p, miR-361-5p, miR-383-5p, miR-411, miR-424-5p, miR-433, miR-545, miR-573, miR-874, miR-1284, miR-2861, and miR-3941. Conversely, seven microRNAs were upregulated in cervical cancer: miR-31, miR-92a, miR-93, miR-96-5p, miR-199b-5p, miR-200a, and miR-224. Certain dysregulated microRNAs showed stage-specific associations with the progression of cervical cancer. These differentially expressed microRNAs are implicated in critical cancer hallmarks, such as evading growth suppressors, enabling replicative immortality, activating invasion and metastasis, resisting cell death, and sustaining proliferative signaling. The study also suggests that specific stage-associated microRNAs could serve as biomarkers for classifying and monitoring the progression of cervical cancer.

A piece of research aimed to pinpoint microRNAs that could predict how patients with locally advanced cervical cancer would respond to treatment [45]. The study enrolled 41 patients and used a miScript microRNA PCR array for profiling. The analysis revealed 101 microRNAs with significant differential expression between those who fully responded to treatment and those who did not. Specifically, miR-31-3p, miR-144-3p, and miR-3176 were found to be more abundant in non-responders, while 97 other microRNAs were less abundant. To confirm these findings, seven microRNAs were chosen for validation using RT-qPCR. A signature consisting of miR-31-3p, miR-100-5p, miR-125a-5p, miR-125b-5p, miR-200a-5p, miR-342, and miR-3676 was identified as significantly linked to clinical response. This seven-microRNA signature shows promise as a potential biomarker for resistance to radiation and chemotherapy.

### 2.2. Genomic Signature of microRNAs Circulating

A study investigated serum microRNA alterations in cervical cancer patients to identify early diagnostic markers [46]. The researchers analyzed serum samples from 213 cervical cancer patients and 158 control subjects. MicroRNA expression was initially screened using Solexa sequencing, with validation through RT-qPCR. This analysis revealed 12 microRNAs that were notably elevated in the serum of cervical cancer patients compared to the control group, including miR-21, miR-25, and miR-29a. A specific panel of five microRNAs (miR-21, miR-25, miR-29a, miR-200a, and miR-486-5p) showed superior diagnostic capability in terms of sensitivity and specificity when compared to individual microRNA tests. These five microRNAs were subsequently confirmed as biomarkers for cervical cancer. Furthermore, miR-29a and miR-200a were significantly upregulated in poorly differentiated tumors, indicating their potential utility in disease monitoring and prognosis. The findings suggest that this five-microRNA signature, identified through comprehensive serum microRNA profiling, could serve as a non-invasive diagnostic tool for cervical cancer.

Another study has identified a promising set of circulating microRNAs that could serve as non-invasive biomarkers for cervical cancer diagnosis. Researchers developed a four-microRNA signature (miR-16-2*, miR-195, miR-497, and miR-2861) capable of distinguishing cervical cancer patients from individuals with CIN and healthy controls. The research involved a cohort of 184 cervical cancer patients, 186 CIN patients, and 193 healthy subjects from whom serum samples were collected. Initially, 444 microRNAs were screened using RT-qPCR, narrowing down to 66 in the training phase and finally to 7 in the validation phase. The expression levels of miR-16-2* and miR-497 were found to be elevated in the serum of cervical cancer patients, while miR-195 and miR-2861 showed decreased expression. This microRNA panel demonstrated high accuracy in differentiating between the groups. The findings suggest that this microRNA panel could be a valuable tool for cervical cancer diagnosis, offering high sensitivity and specificity. This builds upon previous work, including contributions from Zhang et al., in establishing serum microRNA signatures for cervical cancer diagnosis [47].

A study by Xin et al. investigated the potential of circulating microRNAs as biomarkers for the early detection of CIN [48]. The research involved a cohort of 126 CIN patients and 60 healthy control individuals. Using RT-qPCR, they measured the expression levels of a specific microRNA panel in serum samples from all participants. The results showed a significant upregulation of four microRNAs (miR-9, miR-10a, miR-20a, and miR-196a) in the serum of CIN patients compared with the control group. Furthermore, a strong correlation was observed between HPV infection status and the expression levels of these microRNAs. This four-microRNA signature proved to be highly effective in distinguishing individuals with CIN from healthy controls. These findings suggest that these four microRNAs may act as oncogenes in cervical cancer, and their serum levels could serve as indicators for the progression of CIN and cervical cancer.

Recent research has explored the use of a cell-based assay to identify unique transcriptional patterns in peripheral blood mononuclear cell (PBMC) cultures driven by serum from individuals with cervical cancer [49]. This method involved exposing healthy PBMC reporters to serum samples from women with either localized or metastatic cervical cancer and then analyzing the resulting gene expression changes across the entire genome. Through computational analysis, four specific microRNAs (miR-23a-3p, miR-23b-3p, miR-193b-5p, and miR-944) were pinpointed as potential regulators, given their known ability to target and suppress the expression of multiple genes that showed differential activity. Out of these four, two (miR-23a-3p and miR-944) were subsequently confirmed using RT-qPCR in a separate group of women diagnosed with local or metastatic cervical cancer. These findings suggest that a signature based on these four microRNAs, identifiable through the transcriptional changes induced by patient serum in PBMC cultures, could potentially help distinguish between localized and metastatic forms of cervical cancer.

Research investigated the potential of microRNAs as non-invasive biomarkers for the early detection and prognosis of cervical cancer [50]. The study analyzed the expression of six specific miRNAs, including three oncomiRs (miR-21, miR-199a, and miR-155-5p) and three tumor suppressor microRNAs (miR-34a, miR-145, and miR-218), in samples from healthy individuals, pre-cancer patients, and cancer patients. These samples included urine, serum, cervical scrapes, and tumor tissue. The findings indicated that a specific combination of urine microRNAs (miR-145-5p, miR-218-5p, and miR-34a-5p) was highly effective in differentiating between healthy controls and those with pre-cancer or cancer, showing 100% sensitivity and 92.8% specificity. This urine microRNA panel also demonstrated a strong correlation with miRNA profiles found in serum and tumor tissue. Additionally, the expression levels of microR-34a-5p and miR-218-5p were identified as independent prognostic indicators for the overall survival of cervical cancer patients.

A study by Ferrero et al. investigated microRNA patterns and expression levels in four different biofluids: plasma-derived exosomes, urine, cervical scrapes, and stool samples [51]. These biofluids were chosen as they represent potential surrogate tissues for diagnostic and screening purposes. The researchers utilized small RNA- sequencing to analyze samples from healthy individuals, specifically 125 plasma-derived exosomes, 48 urine samples, 31 cervical scrapes, and 39 stool samples. The analysis revealed a total of 231 microRNAs that were unique across all specimen types. However, 11 microRNAs were consistently found in all types of samples: miR-320a, miR-589-5p, miR-636, miR-1273a, miR-3960, miR-4419a, miR-4497, miR-4709-5p, miR-4792, miR-7641-1, and miR-7641-2. Plasma exosome samples contained the highest number of specimen-specific microRNAs, with 155 identified. Stool samples followed with 55 specific microRNAs, then urine samples with 22, and cervical scrape samples with only one. Among these specimen-specific microRNAs, miR-122-5p was most abundant in plasma exosome samples. In other sample types, miR-204-5p was most abundant in stool, miR-655-5p in urine, and miR-4741 in cervical scrapes. The identification of these microRNAs through small RNA-sequencing contributes to understanding the human miRNome’s composition in various biospecimens from healthy individuals. This knowledge holds potential for future applications, particularly in cancer screening programs.

In another study, researchers investigated the potential of a specific microRNA signature found in plasma as a diagnostic tool for cervical cancer [52]. The study utilized plasma samples from 97 patients with cervical cancer and 87 healthy controls to identify microRNAs that were dysregulated. The diagnostic utility of both individual microRNAs and panels of microRNAs was then assessed based on their sensitivity and specificity for cervical cancer detection. Further analysis involved examining the expression levels of these microRNAs in plasma exosomes and tissue samples from cervical cancer patients. The study identified four particular plasma microRNAs (miR-21-5p, miR-146a-5p, miR-151a-3p, and miR-2110), which were found to be upregulated in cervical cancer patients. These four microRNAs were subsequently combined to form a diagnostic panel for cervical cancer. It was observed that miR-21-5p and miR-146a-5p were also upregulated in cervical cancer tissue samples, while miR-146a-5p, miR-151a-3p, and miR-2110 showed upregulation in plasma exosomes. These findings suggest that this four-microRNA signature in peripheral plasma could serve as a promising new biomarker for diagnosing cervical cancer.

A study investigated the potential of a microRNA signature for diagnosing and predicting outcomes in early-stage cervical cancer [53]. The research involved 112 patients with early cervical cancer, 45 with CIN, and 90 healthy individuals. Findings revealed that miR-21 levels were elevated, while miR-125b and miR-370 levels were reduced in cervical cancer patients compared to healthy controls, a trend observed in both the GEO GSE30656 and TCGA datasets. The diagnostic and prognostic utility of these microRNAs for early-stage cervical cancer was further assessed. Specifically, in early-stage cervical cancer, serum miR-21 was higher, and serum miR-125b and miR-370 were lower when compared to CIN patients and healthy subjects. The combination of these microRNAs demonstrated good accuracy in distinguishing early-stage cervical cancer from CIN or healthy individuals. Furthermore, a strong positive correlation was found between serum miR-21 and lymph node metastasis and recurrence in early-stage cervical cancer cases. Conversely, serum miR-125b and miR-370 showed negative correlations with lymph node metastasis and recurrence. These findings suggest that a combined panel of serum miR-21, miR-125b, and miR-370 could serve as a valuable tool for detecting and predicting the prognosis of early-stage cervical cancer.

A study by Cao et al. previously identified a three-microRNA serum signature for cervical cancer diagnosis [54]. In a separate analysis, 29 candidate microRNAs were initially identified and validated. The researchers also included four microRNAs (miR-20a-5p, miR-21-5p, miR-196a-5p, and miR-218) that had been previously reported in the literature. Significantly, serum levels of miR-20a-5p and miR-122-5p were found to be upregulated in cervical cancer patients compared to healthy controls, while miR-133a-3p was downregulated. These increased levels of miR-20a-5p and miR-122-5p were consistent across both early and advanced stages of cervical cancer. Exosomal expression of miR-20a-5p and miR-122-5p was also elevated in exosome samples from cervical cancer patients. Subgroup analyses were performed to investigate the relationship between the identified microRNA signature and clinicopathological parameters such as tumor node metastasis stage and histological type. It was observed that the three-microRNA signature did not show specificity for histological subtype, as there was no differential expression between squamous cell carcinoma and adenocarcinoma patients. Overall, these findings suggest that serum miR-20a-5p, miR-122-5p, and miR-133a-3p could serve as useful diagnostic biomarkers for cervical cancer.

The microRNAs with evidence of roles in the genomic signature of cervical cancer are summarized in Table 1.

## 3. Regulatory Genetic Networks Modulated by MicroRNAs and Their Target Genes in Cervical Cancer

Efforts have been dedicated to elucidating the precise mechanisms of carcinogenesis and the identification of involved genes. The regulatory functions of microRNAs on target genes and their roles in carcinogenesis have been the subject of considerable research. MicroRNAs can establish feedback or feedforward loops, which are critical in the regulation of target genes within various biological processes. Disruption of the regulatory balance between microRNAs and their target genes may contribute to cancer development.

Recent research highlights the significant involvement of microRNAs in cervical cancer, particularly in regulating their target genes. Comprehensive analyses of miRNA and mRNA expression profiles and gene regulation networks are crucial for identifying molecular markers and key genes involved in cervical tumorigenesis. One study utilized microarrays to compare microRNA and mRNA expression in normal and cancerous cervical tissues, leading to the construction of a microRNA-gene network [55]. The TargetScan 5.0 database was used to predict microRNA target genes, and these predictions were then cross-referenced with differentially expressed mRNAs. A negative correlation between the intersection and the microRNAs was also analyzed. Bioinformatic methods were employed to assess the functions and pathways of target genes and to build microRNA-gene networks. The findings indicated differential expression of 29 microRNAs and 2036 mRNAs when comparing normal and cervical tumor tissues. Specifically, 13 microRNAs and 754 mRNAs showed upregulation in tumor tissues, while 16 microRNAs and 1282 mRNAs were downregulated. Genes regulated by downregulated microRNAs were implicated in 415 functions, and those regulated by upregulated microRNAs were involved in 163 functions. These gene sets participated in 37 and 17 significant pathways, respectively. The constructed microRNA-gene network identified miR-15a, miR-20b, and miR-106b as miRNAs that regulate the highest number of target genes. These microRNAs are central to the network, exhibiting strong regulatory effects on differential gene expression in cervical cancer. This suggests that the differentially expressed microRNA-gene network plays a critical role in biological functions and signal transduction during cervical carcinogenesis.

To develop new therapeutic approaches for cervical cancer, researchers have analyzed the dysregulation of microRNAs and their target mRNAs [56]. This involved identifying microRNAs that are abnormally expressed in cervical cancer and then predicting their mRNA targets using various databases like TargetScan v7.2, PicTar - Target Prediction v4, and miRanda v3.3a. A key step was to compare these predicted targets with existing gene datasets known to be involved in cervical cancer, resulting in a refined set of putative target genes. The structural characteristics of this putative network, along with validated and human PPI networks, were then examined. Through this analysis, certain genes, including BIRC5 (survivin), HOXA1, and RARB, were identified as highly relevant in cervical cancer due to their topological properties and high scores in databases like Genecards. These genes play crucial roles: BIRC5 is an anti-apoptotic protein, while HOXA1 and RARB are transcription factors that influence cell cycle and apoptosis-related proteins. The network analysis also suggested that miR-30b and miR-203 might regulate these important genes. This research highlights BIRC5, HOXA1, and RARB as significant therapeutic targets in cervical cancer due to their critical involvement in its development.

A study by Wang et al. explored the regulatory genetic networks in cervical cancer [57]. They developed three distinct networks: a differentially expressed network, a related network, and a global network, each built from databases of microRNAs and associated genes. The first network, focusing on differentially expressed elements, identified seven genes and ten microRNAs as experimentally validated in cervical cancer. A key finding was a sub-network centered on PTEN, where TWIST1, miR-214, PTEN, and miR-21 formed a sequential control chain. Notably, miR-21 targets PTEN, creating a self-regulating feedback loop that maintains balance with STAT3 and miR-21. Another part of this network highlighted the joint regulation of miR-143 by TP53 and TGF-B1. The second network, which was larger and more complex, featured prominent microRNAs like miR-21, miR-23b, miR-34, miR-143, and let-7c, alongside transcription factors such as TP53, TP63, and c-Myc. This network also revealed two additional self-adapting feedback mechanisms involving let-7c, miR-34, and c-Myc. Overexpression of c-Myc was observed to diminish the suppressive function of let-7c, and the tumor suppressor miR-34 displayed a local balance adjustment system with c-Myc. The third network investigated interactions among previously identified transcription factors and their 1000-nucleotide base sequences. These transcription factors were integrated with host genes and differentially expressed microRNAs to expand the cervical cancer network system. A self-adapting feedback loop was noted between NF-kB and miR-21, with NF-kB regulating miR-21, miR-214, and let-7b. MiR-21 and miR-214 were identified as core biological factors, emphasizing their importance in the cervical cancer network. Furthermore, NF-kB was found to participate in the sub-system centered on PTEN and miR-21, mediated by miR-21 and miR-214. These findings suggest that this analytical approach could be valuable for uncovering additional core factors, parallel networks, and relevant motifs in the development of cervical cancer and other tumorigenesis processes.

A research team led by Mo et al. developed a sophisticated computational tool called SIG++ (Stochastic process model for Identifying differentially co-expressed Gene pair) to analyze changes in microRNA regulation during the progression of CIN [58]. They conducted transcriptome analysis on microRNAs and mRNAs from 24 cervical samples, representing normal, CIN I, and CIN III stages. The SIG++ algorithm was central to identifying microRNA-mRNA pairs exhibiting significant regulatory shifts. The researchers also defined key concepts such as “efficient pair” and “efficient target” to facilitate the construction of these differential regulation networks. Using human GO biological processes, they built microRNA differential regulatory networks to trace the disease’s progression from normal tissue to CIN I and then from CIN I to CIN III, pinpointing efficient targets and their associated effector biological processes. The findings from this study revealed interesting insights into the pathway enrichment of microRNA regulation. Inflammation was found to be a pathway enriched by stage-specific microRNA-mRNA regulations. In the normal stage, microRNA regulations were particularly enriched in processes like sodium ion transport and glycolysis. As the disease progressed to CIN I, microRNA regulations showed a strong enrichment in cell migration and differentiation. By the CIN III stage, specific microRNA regulations were notably enriched in virus integration. This research introduced the concept of differential regulation by microRNAs and provided a novel algorithm for its identification across different disease stages. The constructed microRNA differential regulation networks offer valuable biological insights. Ultimately, these discoveries highlight how understanding regulatory genetic networks can significantly enhance our comprehension of CIN progression.

A comprehensive meta-analysis conducted by He et al. identified 63 differentially expressed microRNAs (DEmiRNAs) across various stages of cervical cancer development, from CIN I to III and invasive cervical cancer [59]. This extensive study synthesized data from 85 published reports, encompassing a total of 3,922 cases of CIN or cancer and 2099 non-cancerous control tissue samples. The findings of this research highlight that these deregulated microRNAs are functionally involved in crucial cancer-related pathways. These pathways include but are not limited to, the cell cycle, p53 signaling, and Wnt signaling pathways, all of which are well-known to be critical in oncogenesis. To further elucidate the mechanisms by which microRNAs influence cervical cancer pathogenesis, a microRNA-mRNA interaction network was constructed. This network integrated various components, including viral oncoproteins, the identified dysregulated microRNAs, and their predicted and validated target genes. The construction of such a network helps to visualize the complex interplay between these molecular players and their impact on disease progression. The implications of these findings are substantial, suggesting that stage-specific microRNAs hold promise as potential biomarkers. These biomarkers could be utilized for the classification of cancer and for monitoring the progression of the disease, offering new avenues for diagnosis and personalized treatment strategies.

One research study explored the complex interplay of biological processes within cells by analyzing cervical cancer using three key biomolecular networks: PPI, metabolic, and post-transcriptional regulatory networks [60]. This approach allowed for a comprehensive understanding of how signaling, regulatory, and metabolic pathways are interconnected. The study involved a meta-analysis of transcriptomic datasets related to cervical cancer, incorporating data from five distinct studies and a total of 236 samples. Through statistical analyses, researchers identified differentially expressed genes (DEGs) that were central to the disease. Further investigation involved gene set over-representation analyses on these core DEGs, which helped pinpoint significantly enriched pathways and GO. The researchers then validated the differential expression profiles of various reporter biomolecules using independent RNA-Seq and microRNA-Seq datasets. They also successfully demonstrated the prognostic value of several of these biomolecules. The findings of this study offer potential diagnostic or prognostic biomarkers and therapeutic targets for cervical cancer, paving the way for future experimental and clinical research.

An approach for analyzing microRNA-mRNA interactions in cervical cancer, called RFCM3, has been developed [61]. This algorithm aims to uncover significant regulatory modules by integrating microRNA and mRNA expression data. RFCM3 works by identifying microRNA-mRNA modules that maximize the relationship between mRNAs and microRNAs while also considering the functional similarity of the selected mRNAs. The method then combines different modules, utilizing information from the microRNA-microRNA synergistic network, to create broader modules containing multiple microRNAs and mRNAs. The effectiveness of RFCM3 in cervical cancer studies has been demonstrated through enrichment analyses and other standard metrics, showing its superiority over existing methods. The RFCM3 module has been shown to produce more robust, integrated, and functionally enriched microRNA-mRNA modules. This suggests that RFCM3 is a valuable tool for investigating complex biological pathways involving multiple microRNAs and mRNAs in cervical cancer.

Understanding the complex interplay between microRNAs and mRNAs is crucial for deciphering disease mechanisms, especially in cancers like cervical cancer. A bioinformatics study, for instance, delved into the microRNA-mRNA regulatory network in cervical cancer to identify potential prognostic markers [62]. The research began by obtaining microRNA and mRNA expression profiles from the TCGA database. Using the “EdgeR” package, a significant number of differentially expressed microRNAs (DEmiRNAs) and mRNAs (DEmRNAs) were identified when comparing healthy and cancerous cervical tissues. Specifically, 5096 DEmRNAs and 114 DEmiRNAs were found. Further analysis pinpointed 102 target DEmRNAs that were regulated by upregulated DEmiRNAs and 150 target DEmRNAs associated with downregulated DEmiRNAs. To understand the broader molecular context, a PPI network was constructed using Cytoscape software v3.10.3. This allowed for the identification of “hub genes,” which are central to the network’s function. These hub genes were then used to build a microRNA-hub gene network. The network analysis revealed 10 top upregulated and 10 top-downregulated hub genes. Interestingly, certain DEmiRNAs, such as miR-23b-3p, miR-106b-5p, and miR-200c-3p, were indicated as potential modulators of several hub genes. Pathway enrichment analysis showed that upregulated DEmiRNAs were linked to cancer-related pathways like proteoglycans in cancer and focal adhesion, while downregulated DEmiRNAs were associated with pathways in cancer, cell cycle, and viral carcinogenesis. To validate the prognostic significance of the identified DEmiRNAs, their roles in cervical cancer were assessed using the “survival” package. This validation step identified several DEmiRNAs associated with cervical cancer prognosis: four upregulated DEmiRNAs (miR-106b-5p, miR-200c-3p, miR-210-5p, and miR-425-5p) and thirteen downregulated DEmiRNAs (miR-23b-3p, miR-29c-5p, miR-99a-5p, miR-101-3p, miR-143-5p, miR-145-3p, miR-145-5p, miR-181c-5p, miR-502-5p, miR-504-5p, miR-505-5p, miR-532-5p, and miR-6507-5p). Beyond individual genes and miRNAs, the study also developed a prognostic signature based on three key hub genes: EZH2, FLT1, and GAPDH. This prognostic signature’s value was further evaluated using a nomogram. A notable finding was the relationship between these three genes and the infiltration of various immune cells, including dendritic cells, M0 and M1 macrophages, mast cells, and monocytes, as investigated using the CIBERSORT algorithm. This suggests a potential link between these genes and the immune response within the tumor microenvironment. Thus, this comprehensive bioinformatics analysis provides a deeper understanding of the microRNA-mRNA networks in cervical cancer progression and proposes a valuable prognostic gene signature for patients with cervical cancer.

The investigation of cervical cancer at a molecular level has involved extensive bioinformatics analyses, combining microRNA profiling with gene expression data. Various computational approaches, such as differential expression analysis, PPI networks, and ceRNA network analysis, have been applied to pinpoint microRNAs, genes, and relevant pathways in cervical carcinogenesis. A key aspect of this research is the role of lncRNAs, which are RNA molecules over 200 nucleotides long that do not code for proteins but regulate gene expression epigenetically and transcriptionally [63]. These studies have led to the construction of complex regulatory networks, including those of differentially expressed microRNAs (DEmiRNAs) and genes (DEGs), PPI networks, and transcription factor-target networks. Further analysis integrated lncRNAs associated with DEmiRNAs to build lncRNA-microRNA-target-ceRNA networks. Through these analyses, 18 DEmiRNAs and 620 DEGs were identified. The DEmiRNAs showed enrichment in 35 KEGG pathways, notably the PI3K-Akt signaling pathway, which involves miR-451a. DEGs were linked to various functions, such as DNA replication involving E2F7 and angiogenesis involving EREG. Key elements identified in these networks include the DEmiRNA-DEGs regulatory network, which contained 120 nodes and 216 interaction pairs, with miR-106b-5p having the highest connectivity; regulation of EREG by miR-148a-3p and E2F7 by miR-451a; E2F7′s involvement in the transcription factor-target regulatory network, where it regulates CDC6; and a ceRNA network composed of 15 lncRNAs, 11 microRNAs, and 90 DEGs. Within this network, miR-148a-3p was found to interact with lncRNA HOTAIR. These findings highlight E2F7, EREG, miR-106b-5p, and miR-451a as critical genetic factors influencing the progression of cervical cancer.

An investigation into microRNA-mRNA networks in cervical cancer utilized microarray datasets from the GEO database [64]. Analysis of DEGs using the Robust Rank Aggreg R package revealed 209 DEGs, with 115 upregulated and 94 downregulated. GO enrichment and KEGG pathway analyses were performed to understand the functions and pathways of these DEGs. Candidate differentially expressed microRNAs (DEmiRNAs) were identified, including nine upregulated microRNAs (let-7c-5p, miR-10b-5p, miR-101-3p, miR-101-5p, miR-195-5p, mir-204-5p, miR-377-5p, miR-497-5p, and mir-6507-5p) and twelve downregulated microRNAs (miR-7-5p, miR-15b-5p, miR-16-2-3p, miR-18a-3p, miR-18a-5p, miR-31-5p, miR-130b-3p, miR-130b-5p, miR-183-5p, miR-203a-3p, miR-3934-5p, and miR-4638-3p). A regulatory network was then constructed based on the negative regulation between these DEmiRNAs and their target genes. Within this network, two microRNAs, mir-101-3p and mir-6507-5p, were identified as hub DEmiRNAs linked to the prognosis of cervical cancer patients. Furthermore, a five-gene signature (APOBEC3B, DSG2, CXCL8, ABCA8, and PLAGL1) was chosen through prognosis signature analysis and may serve as an independent prognostic indicator for stratifying risk subgroups in cervical cancer patients. The risk score of this prognostic model was also found to correlate with immune cell infiltration, specifically with mast cell activation, resting natural killer cells, resting dendritic cells, T regulatory cells (Tregs), and T follicular helper cells. The Connectivity Map database was screened for small molecule drugs based on the DEGs, leading to the identification of three potential new treatments for cervical cancer: thioguanosine, apigenin, and trichostatin A. These findings underscore the clinical significance of the microRNA-mRNA regulatory network and the prognostic model in the context of cervical cancer prognosis and treatment.

## 4. MicroRNA-mRNA Regulatory Modules in Cervical Cancer

The regulatory role of microRNAs in gene expression has garnered significant attention. The identification of microRNA-mRNA regulatory modules is crucial for deciphering the aberrant transcriptional regulatory networks in cervical cancer. While representing a bioinformatics challenge, this endeavor will enhance the understanding of the role of known biomarkers, such as microRNAs and mRNAs, in various pathways in cervical cancer.

Different methodologies exist for analyzing the intricate relationships within microRNA-mRNA networks [65]. While correlation-based methods have been used, they often operate under the inaccurate assumption that a single microRNA exclusively targets a single mRNA. To overcome this limitation, linear modeling approaches have been developed [66]. These models treat mRNA expression as a linear combination of microRNAs and employ Bayesian algorithms to identify hidden microRNA targets [67]. Furthermore, linear modeling techniques can incorporate various distribution methods, integrating pre-existing sequence and structural data. Another effective strategy for integrating microRNA and mRNA interactions is the Bayesian network technique [68]. This method facilitates an integrated analysis of differentially expressed microRNAs and mRNAs. Given the vast amount of biological data now available, developing scalable solutions is crucial, and machine learning models based on Bayesian networks offer a promising approach [69].

In biological systems, all occurrences are governed by a specific organizational structure. This fundamental concept has spurred advancements in statistical methodologies for integrating microRNA and mRNA data. Researchers have employed statistical approaches, independent of prior knowledge, to analyze diverse microRNA-mRNA expression datasets and establish a method for distinguishing between different tissue types. Similarly, Nersisyan et al. introduced a new tool for constructing microRNA-gene-transcription factor networks using a comparable strategy [70]. An alternative approach for developing microRNA-mRNA networks involves probability learning-based techniques. This method estimates the interaction probabilities of known microRNA-mRNA pairs [71]. However, the successful and robust implementation of this operation necessitates the incorporation of various information sources. Another significant machine learning technique is non-negative matrix factorization. This technique involves a joint analysis of microRNA and gene expression profiles from multiple information sources, alongside the simultaneous integration of additional network data, to generate meaningful microRNA-mRNA groupings [72].

Identifying functional relationships between microRNAs and mRNAs is crucial in biological research. Various computational platforms and methodologies have been developed to achieve this [73]. Gamberger et al. employed the CN2-SD system as the rule generation system to identify microRNA-mRNA groups [74]. The primary characteristic of this tool is its microRNA-mRNA grouping methodology. Other methods leverage statistical or correlation information to establish these groupings. For instance, platforms like miRcorrNet utilize correlation data to generate microRNA-mRNA groups. Similarly, techniques such as MiR Module Net integrate statistical information to form these associations [75]. The goal of these diverse strategies is to uncover significant microRNA-mRNA interactions that can provide insights into gene regulation and disease mechanisms.

The miRNet (https://www.mirnet.ca/) (14 April 2021 accessed), is a valuable web-based platform designed to facilitate the elucidation of microRNA functions through the integration of user data with existing knowledge via network-based visual analytics [76]. The miRNet version 2.0 addresses evolving bioinformatics needs and challenges by enabling the creation of intricate microRNA-centric networks for systems-level interpretation of microRNA functions and gene regulation. To investigate the regulatory pathways of microRNA-gene networks in cervical cancer, we utilized the features of the miRNet 2.0 website platform. Data from cervical cancer (Table 1) were employed to determine the role of the oncomiR miR-21-5p in cervical cancer pathogenesis. In this analysis, microRNA target analysis was conducted by searching relevant microRNA-gene networks despite the platform’s additional functionalities, such as a transcription factor-gene database.

The miRNet 2.0 platform was utilized to reconstruct and visualize a microRNA-gene regulatory network focusing on miR-21-5p, identified as a highly expressed microRNA in cervical cancer. This network, built within the microRNA module, initially contained one microRNA node and 612 gene nodes connected by 612 edges. A degree cutoff of 1.0 was applied to improve its visual clarity. The resulting network displays microRNAs in a central zone, with target genes in an intermediate layer organized by degree of centrality. MiRNet 2.0 was able to confirm key targets based on their degree measures. The platform also facilitated functional enrichment and module analysis of the complete network. Through this analysis, additional networks with alternative pathways, such as the MAPK signaling pathway, were identified after manually adjusting edge thickness. Functional enrichment analysis using the KEGG database highlighted the MAPK signaling pathway, which included 25 potential target genes regulated by miR-21-5p, such as AKT2, ATF2, DAXX, DUSP8, DUSP10, EGFR, FAS, FASLG, FGF12, IL1B, MEF2C, MAP3K1, MAP2K3, MAP3K2, MKNK2, MYC, NF-KB1, NTF3, RASA1, RASGRP1, RASGRP3, RPS6KA3, TGF-B1, TGF-B2, and TGFBR2. Specifically, MAP3K1, a component of the MAPK signaling pathway, was identified as a bioinformatic target of miR-21-5p. While MAP3K1 is a serine/threonine kinase involved in signal transduction cascades like the ERK, JNK, and NF-kappa-B pathways, all implicated in carcinogenesis [77], its regulation by miR-21-5p has not been experimentally confirmed in humans according to NCBI. This process demonstrates the platform’s capability to generate comprehensive regulatory networks and identify significant modules for detailed study, potentially uncovering previously unconfirmed microRNA target genes. As shown in Figure 2, the microRNA-gene regulatory network is displayed at the center of the network viewer page in “default layout”. It illustrates several interactions between microRNAs (center zone) and target genes (middle layer).

## 5. Gene Therapy Clinical Trials with MicroRNAs for Uterine Cancers

New avenues for treating cervical cancer are emerging, particularly for advanced or recurrent cases where conventional therapies like radiation and chemotherapy often yield limited and short-lived responses. While early detection and treatment of precancerous lesions are crucial, options for metastatic or recurrent disease remain restricted. A promising strategy involves gene therapy that targets HPV components, as HPV plays a significant role in carcinogenesis in cervical and other tumors. The HPV E6 and E7 oncoproteins are known to interact with various cellular proteins in both the nucleus and cytoplasm, thereby influencing molecular pathways that contribute to tumor development. Consequently, researchers are exploring methods to block the expression of these viral oncoproteins. Current research focuses on developing HPV therapeutic vaccines. These include approaches utilizing viral vectors such as *Herpes simplex* virus, adenoviral and adeno-associated viral vectors, lentiviral vectors, and measles virus. Non-viral delivery systems are also under investigation, including shRNA, siRNA, microRNA, naked plasmid DNA-based vaccines, lipid-based and polymeric nanoparticles, CRISPR/Cas9, HPV peptides and proteins, and specific tumor and dendritic cells. Lentiviral vectors have been specifically used in some of these strategies for delivering target genes [78,79].

The current gold standard for treating advanced or metastatic cervical cancer involves a combination of cisplatin-based chemotherapy and radiotherapy. This dual approach has shown improved patient outcomes compared with chemotherapy alone or when combined with hysterectomy [80,81]. The effectiveness of cisplatin stems from its ability to induce DNA damage and activate DNA repair mechanisms, while radiotherapy utilizes ionizing radiation to harm undesirable cells. Both therapies aim to precisely target the tumor with a specific dose, striving for cancer cell elimination while minimizing harm to healthy surrounding tissues. However, a major hurdle in treatment success is the development of resistance to chemotherapy and/or radiotherapy. This resistance can be either present from the outset (primary) or develop during treatment (acquired), leading to systemic toxicity, adverse effects, and ultimately, disease recurrence, progression, and higher mortality rates. Despite efforts to standardize treatment doses across patients and the use of advanced techniques, local recurrences remain a common challenge. This underscores the need for new strategies to overcome resistance and improve the long-term prognosis for patients with cervical cancer [82,83].

The medical community is increasingly recognizing the importance of personalized medicine across various clinical fields. In the realm of cancer treatment, microRNAs are emerging as significant regulators of how cells react to therapies such as chemotherapy and radiation. Their influence on drug and radiation sensitivity or resistance is multifaceted, involving mechanisms like changes in DNA repair, modulation of cell cycle checkpoints, and alterations in the tumor’s immediate environment and programmed cell death. Given their role, microRNAs are considered promising indicators for predicting patient response to chemotherapy or radiation and for developing tailored treatment plans. This could pave the way for new, individualized interventions in cancer care. For cervical cancer specifically, the identification of novel, effective biomarkers and genetic signatures is crucial for more precise diagnosis, prognosis, and the selection of optimal treatment strategies. The recent progress in microRNA-targeted therapies also presents new opportunities to devise strategies that can modify a patient’s tumor biology and response to treatment, thereby facilitating the development of new targeted approaches for cervical cancer therapy.

Table 2 provides a summary of information pertaining to uterine cancer microRNA gene therapy clinical trials worldwide. The data presented were compiled from the official clinical trials database of the National Library of Medicine, National Institutes of Health, United States [84]. Many clinical trials have incorporated evaluations of safety and tolerability in conjunction with gene therapy strategies. Several studies are ongoing or in the participant recruitment phase, while others have been completed.

In the ClinicalTrials.gov identifier NCT04087785, researchers analyzed microRNA expression in tissue samples from early-stage cervical cancer patients who underwent radical hysterectomy and lymphadenectomy. These samples were categorized based on the presence or absence of lymph node metastasis for microRNA profiling. The profiling utilized the GeneChip miRNA 3.0 array from Affymetrix to identify global microRNA profiles. The study aimed to identify microRNAs with prognostic value in cervical cancer patients with positive lymph node metastasis. Differential expression of microRNAs was determined using a cutoff value of *p* < 0.01 and a fold change of 1.5, classifying them as either over-expressed or under-expressed. The findings from this study have not yet been published.

The study, registered as NCT03824613 on ClinicalTrials.gov, combines a prospective feasibility design with a retrospective cohort analysis. This investigation focuses on the diagnostic utility of microRNAs in urine for identifying endometrial cancer subtypes. The primary goal is to establish a link between microRNA expression patterns and definitive histological classifications of endometrial cancer. As of now, the findings from this study have not been disseminated through publication.

The ClinicalTrials.gov identifier NCT04845425 is an observational study with a retrospective cohort. This study investigates microRNA expression to find new biomarkers for endometrial cancer, focusing on patient stratification based on molecular alterations as defined by TCGA. The research will also integrate molecular findings with clinical-pathological data. While TCGA has categorized endometrial carcinomas into four prognostic groups based on molecular changes, carcinomas lacking a clear molecular profile show considerable variation in both molecular alterations and biological aggressiveness [85].

This multicenter, open-label, non-randomized study, registered as NCT03776630 on ClinicalTrials.gov, is focused on gynecological cancers. One of its primary objectives is to assess the diagnostic utility of a 5-microRNA index in plasma for determining the risk of lymph node metastasis in endometrial cancer. The study also plans to investigate how this 5-microRNA index correlates with established indicators of lymph node involvement in endometrial cancer. For ovarian cancer, the research aims to confirm prior observations regarding the predictive value of changes in miR-200b plasma concentrations, measured before and after treatment, in relation to progression-free survival. Additionally, a secondary goal of the study is to evaluate multiplexed homogeneous microRNA detection using RCA-FRET against the more traditional RT-qPCR method in plasma samples. As of now, the study’s findings have not yet been published.

This investigation is registered under the ClinicalTrials.gov identifier NCT01119573, where microRNAs are being evaluated as potential biomarkers in tissue samples from patients with either stage I or III endometrial cancer. This study aims to discover specific microRNA expression signatures that correlate with lymph node metastasis in individuals diagnosed with endometrial cancer. Researchers have analyzed banked tumor tissue samples to profile microRNA expression. The methods used for this profiling include microarray analysis and RT-PCR assays. It is worth noting that the findings from this research have not yet been made public.

This study, registered as NCT04010487 on ClinicalTrials.gov, aims to uncover the genetic and molecular pathways driving the malignant transformation of adenomyosis. The research involves analyzing both eutopic endometrium and normal adenomyosis tissue samples from patients diagnosed with endometrial carcinoma arising in adenomyosis (EC-AIA). These tissues are precisely acquired using laser microdissection. To identify the key genes and potential molecular mechanisms of EC-AIA, the study will utilize whole exome sequencing and transcriptomics, specifically RNA sequencing. The project will also compare changes in the expression of different RNA types, including mRNA, microRNA, and lncRNA, between eutopic endometrium, ectopic endometrium, and cancerous tissues through transcriptome sequencing. As of now, the findings from this study have not yet been published.

The ClinicalTrials.gov identifier NCT02983279 aims to explore the effects of short-term calorie restriction on various aspects of cancer. The study focuses on patients with biopsy-confirmed breast, endometrial, or prostate cancers. A key objective is to determine how calorie restriction impacts biomarkers, specifically serum miR-21, which is known to be an oncomiR linked to cancer outcomes. The researchers plan to evaluate changes in miR-21 expression in serum using a two-sided paired t-test. Genomic analysis will involve comparing initial cancer biopsy specimens with definitive surgical specimens. Beyond molecular changes, the trial will also investigate the measurable effects of caloric restriction on both patient and tumor characteristics. This includes assessing the patient’s nutritional status and calculating caloric needs. The study will also track clinical outcomes such as local recurrence, progression-free survival, distant metastases, and overall survival. As of now, the findings from this study have not been published.

The ClinicalTrials.gov identifier NCT05292573 aims to investigate factors influencing the progression of endometrial hyperplasia. This research will enroll 1000 participants diagnosed with simple or complex hyperplasia without atypia. The primary goal is to assess their progression to endometrial cancer, with a particular focus on the potential impact of metformin intervention. This evaluation will be facilitated by linking study data with national health databases maintained by the Health and Welfare Data Science Center in Taiwan. Further molecular analysis will be conducted on patient cohorts, specifically comparing individuals who experience progression to those who do not. This will involve whole genome sequencing to identify genetic variations associated with disease trajectory. Additionally, a microRNA panel will be analyzed in both tissue and serum samples, utilizing existing data (CMRPG3G1511-3) on microRNA profiles and endometrial cancer progression rates. The integration of genomic and microRNA data with clinical outcomes is expected to provide valuable insights into early detection strategies and personalized treatment approaches for endometrial hyperplasia.

The ClinicalTrials.gov identifier NCT03742869 is a multi-omics study of uterine cervical adenocarcinoma patients with and without HPV infections. Peripheral venous blood and samples of cancer tissue and tissue adjacent to cancer will be collected from eligible patients. Alterations in RNA expression patterns, including mRNA, microRNA, and lncRNA, will be compared between patients with and without HPV integration. This research aims to analyze molecular changes in patients with cervical adenocarcinoma, correlating them with HPV infection status. A comprehensive omics approach will be employed, including genomic and transcriptomic analyses, to identify variations in gene expression, such as mRNA, microRNA, and lncRNA. The integration of HPV will be determined through a genome-wide association study. The results of this study have not yet been published.

## 6. Conclusions and Perspectives

Prior investigations have documented microRNAs as potential targets of HPVs or have established associations between specific microRNAs and the process of cervical carcinogenesis. The present study encompasses a comprehensive review of research endeavors that have employed diverse technological platforms for the analysis of microRNAs under a variety of experimental conditions. From these publications, we identified microRNAs that have been reported to exhibit differential regulation in cervical cancer tissues or in vitro model systems. In addition, we conducted an analysis to identify which microRNAs can serve as a genomic signature for patients with cervical cancer. The novelty and interesting thing about this analysis is that it is possible to identify circulating microRNAs, that is, in patients’ biofluids. Therefore, this proposal represents an advantage for patients because it is a non-invasive intervention since the expression profile of microRNAs in different patient biofluids can be identified. These microRNA genomic signatures can be validated by the different integrative multi-omics approach strategies.

While the precise regulatory mechanisms of E6 and E7 viral oncoproteins in HPV-positive cancer cells remain partially understood, particularly concerning their interaction with microRNA-gene networks, research has begun to explore whether changes in cellular microRNA profiles in these cells are linked to the levels of endogenous E6 and E7 expression. Further investigation is needed to fully delineate these complex relationships and their implications for cancer progression and potential therapeutic strategies.

Despite significant strides in medical science, the prognosis for cervical cancer patients remains highly variable. Early detection is a critical factor, as individuals diagnosed in the initial stages often experience favorable outcomes with appropriate treatment. However, patients with advanced-stage cervical cancer face a considerably poorer prognosis. This disparity highlights the urgent need for new strategies to improve patient care. The development of novel biomarkers holds immense promise in this regard. These molecular indicators could revolutionize the way cervical cancer is diagnosed and how its progression is monitored. Furthermore, biomarkers could play a pivotal role in guiding personalized treatment plans, particularly for those with advanced disease, ultimately leading to more effective interventions and improved patient survival rates. The ongoing search for these biomarkers is, therefore, crucial in transforming the landscape of cervical cancer management.

The recent development of microRNA-targeted therapy has opened new avenues of research focused on the identification of novel pharmacological agents capable of modulating tumor biology and therapeutic responsiveness in patients, potentially leading to the development of innovative targeted treatment strategies. A substantial number of microRNAs have been identified as possessing potent tumor-suppressive or oncogenic (oncomiR) functions, with the capacity to regulate multiple cellular pathways and, consequently, potential utility in targeted cancer therapy; however, the available body of pre-clinical and clinical evidence remains limited. A detailed and thorough understanding of microRNA functions may pave the way for the development of novel therapeutic approaches to cervical cancer.

To advance our understanding of cervical cancer, a thorough investigation into changes in microRNA expression and their associated target genes is crucial. This research can shed light on the molecular mechanisms involved in the disease’s onset, progression, diagnosis, and treatment. While individual studies have yielded important results, a definitive grasp of the core molecular drivers of cervical cancer necessitates an integrated multi-omics approach. Integrating biomolecular networks with data from various omic levels through bioinformatics is a key area of focus in regulatory genetic network research. This interdisciplinary approach offers valuable insights into genome reprogramming during disease states. It also helps identify relevant biological entities that could serve as potential diagnostic or therapeutic targets for cervical cancer.

Given the intricate nature of microRNA regulatory networks, understanding the central components within them is crucial for deciphering their roles in biological processes. These networks are characterized by complex interactions, including feedback and feedforward loops, among multiple microRNA targets. The cooperative regulation by microRNAs, particularly in diseases like cancer, is increasingly recognized as a significant area of study. To aid in this research, the miRNet platform offers a user-friendly, web-based tool for analyzing module hubs in microRNA genetic networks. This platform allows users to construct comprehensive regulatory networks, providing a broader perspective on microRNA involvement in various health conditions, including cervical cancer.

## Figures and Tables

**Figure 1 biomedicines-13-01457-f001:**
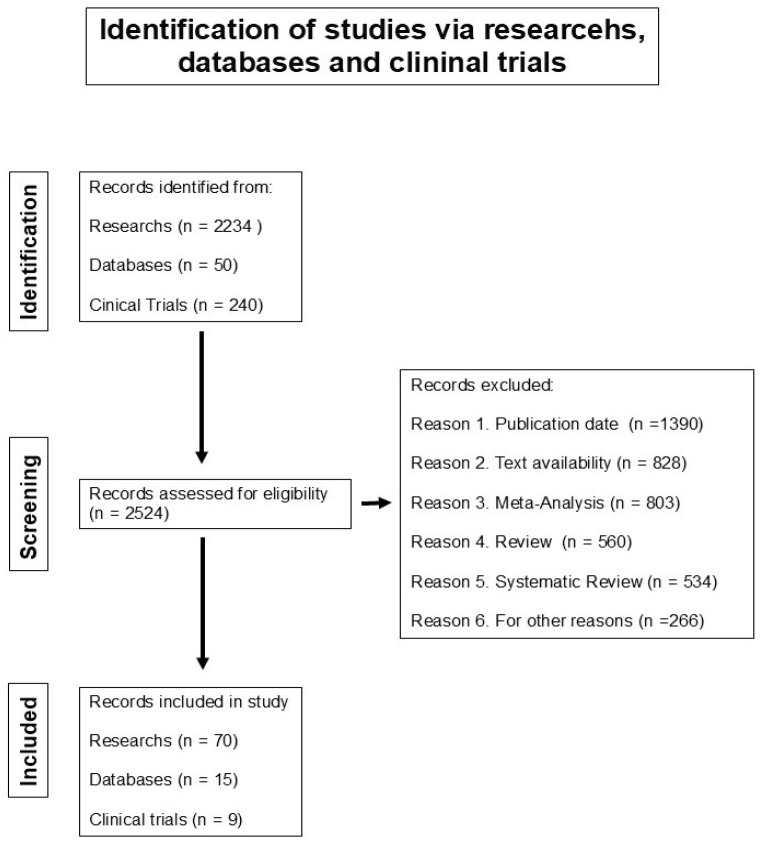
PRISMA FLOW DIAGRAM. Prisma flow diagram, which includes a search of database records, research registers, and clinical trials.

**Figure 2 biomedicines-13-01457-f002:**
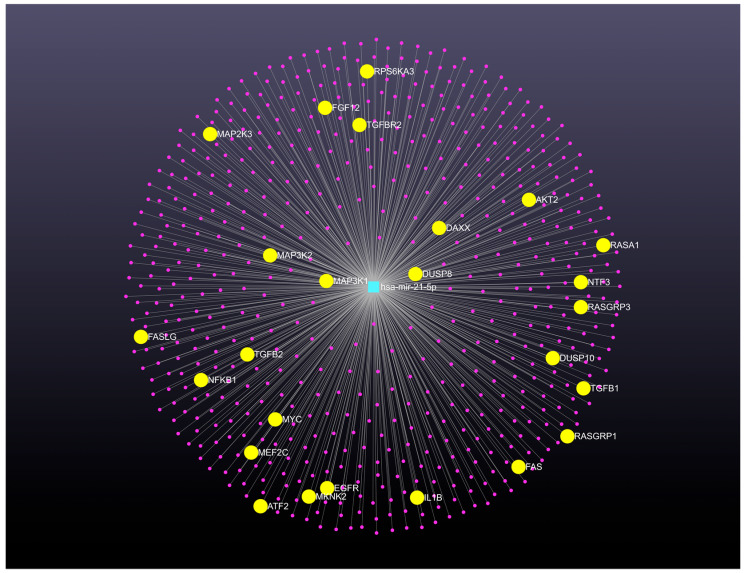
Visualization of the regulatory genetic network of miR-21-5p showing the main features and several network layouts. The central panel shows a network in a circular-tripartite layout, and the surrounding panels provide functions for network analysis and customization. The central panel depicts-miR-21-5p and shows the target genes and 25 hits of the MAPK signaling pathway. The regulatory genetic network of the miR-21-5p was generated using miRNet version 2.0 website platform.

**Table 1 biomedicines-13-01457-t001:** Genomic signature of microRNAs in cervical cancer.

Biospecimen	Genomic Signature of Up-Expressed microRNAs	Genomic Signature of Down-Expressed microRNAs	Reference
Cervical tissues	miR-9, miR-21, miR-200a, miR-203, and miR-218.		[27]
Cervical tissues	miR-26a, miR-29a, miR-99a, miR-143, miR-145, miR-199a, miR-203 and miR-513.	miR-16, miR-27a, miR-106a, miR-142-5p, miR-197 and miR-205.	[28]
Cervical tissues	miR-9, miR-10a, miR-20b, miR-34b, miR-34c, miR-193b, miR-203, miR-338, miR-345, miR-424, miR-512-5p, and miR-518a		[29]
Cervical tissues	miR-15b, miR-16, miR-21, miR-21* miR-188-5p, miR-483-5p, miR-663, miR-765, miR-1300	miR-142-3, miR-149, miR-152, miR-218, miR-374b, miR-376c	[30]
Cervical tissues	miR-1, miR-21		[31]
Cervical tissues	miR-466		[33]
Cervical tissues	let-7f-5p, miR-128-2-5p, miR-130a-3p, miR-202-3p, miR-205-5p, miR-323a-5p, miR-3136-3p	miR-381-3p, miR-4531	[34]
Cervical tissues	miR-216b-5p, miR-585-5p, and miR-7641		[35]
Cervical tissues	miR-145, miR-200c, miR218-1		[36]
Cervical tissues	miR-21-5p, miR-135b-5p, miR-363-3p, and miR-429	miR-136-5p, miR-218-5p, miR-377-3p, miR-497-5p, miR-1184, miR-3196, miR-4687-3p, miR-5587-3p, and miR-5572	[37]
Cervical tissues	miR-20a, miR-92a, miR-141, miR-183*, miR-210, and miR-944		[38]
Cervical tissues	miR-9-5p, miR-15b-5p, and miR-28-5p	miR-100-5p, miR-125b-5p, miR-149-5p, miR-203a-3p, and miR-375	[39]
Cervical tissues	miR-378c and miR-642a		[40]
Cervical tissues	miR-130a	miR-30c, miR-143, miR-372, and miR-375	[41]
Cervical tissues	miR-144, miR-147b, miR-218-2, miR-425, miR-451, miR-483, and miR-486.		[42]
Cervical tissues		miR-34a, miR-200a, and miR-455	[43]
Cervical tissues	miR-31, miR-92a, miR-93, miR-96-5p, miR-199b-5p, miR-200a, and miR-224	miR-1, miR-107, miR-132, miR-139-3p, miR-143, miR-195, miR-335-5p, miR-337-3p, miR-361-5p, miR-383-5p, miR-411, miR-424-5p, miR-433, miR-545, miR-573, miR-874, miR-1284, miR-2861, and miR-3941	[44]
Cervical tissues	miR-31-3p, miR-100-5p, miR-125a-5p, miR-125b-5p, miR-200a-5p, miR-342, and miR-3676		[45]
Circulating	miR-21, miR-25, miR-26b, miR-29a, miR-29c, miR-101, miR-140-3p, miR-146b-5p, miR-191, miR-200a, miR-423-3p, and miR-486-5p		[46]
Circulating	miR-16-2* and miR-497	miR-195 and miR-2861	[47]
Circulating	miR-9, miR-10a, miR-20a, and miR-196a		[48]
Circulating	miR-23a-3p, miR-23b-3p, miR-193b-5p, and miR-944		[49]
Circulating	miR-21, miR-199a, and miR-155-5p		[50]
Circulating	miR-320a, miR-589-5p, miR-636, miR-1273a, miR-3960, miR-4419a, miR-4497, miR-4709-5p, miR-4792, miR-7641-1, and miR-7641-2		[51]
Circulating	miR-21-5p, miR-146a-5p, miR-151a-3p, and miR-2110		[52]
Circulating	miR-21	miR-125b and miR-370	[53]
Circulating	miR-20a-5p and miR-122-5p	miR-133a-3p	[54]

**Table 2 biomedicines-13-01457-t002:** Active microRNA gene therapy clinical trials worldwide to uterine neoplasm.

ClinicalTrials.gov Identifier	Study Title/Country	Condition Diseases and Biospecimen	Study Type, Study Design and Status	Study Population	Current Primary Outcome	First Submitted Date/ Last Update Posted Date
NCT04087785	MicroRNA profile associated with positive lymph node metastasis in early-stage cervical cancer. México	Cervical cancer Samples from formalin-fixed paraffin-embedded tissue blocks.	Observational. Observational, retrospective Completed.	Patients diagnosed with cervical cancer between January 2006 and December 2013 at the Department of Oncologic Gynecology of the National Cancer Institute (Mexico City).	Prognostic miRNAs	11 September 2019 13 September 2019
NCT03824613	Urinary microRNA expression in endometrial cancer patients—a feasibility study United Kingdom	Endometrial Cancer. Urine samples with DNA or microRNA	Observational. Observational, prospective Completed	Endometrial cancer patients	Accuracy of predictive value of miRNA test in detecting endometrial cancer	5 December 2018 1 March 2021
NCT04845425	Identification of miRNAs in endometrial cancer as novel diagnostic and prognostic biomarkers Italy	Endometrial Cancer. Biospecimen is not provided.	Observational. Cohort, retrospective Recruiting	Women with endometrial cancer	Evaluate miRNA expression based on the 4 molecular groups recently identified	10 April 2021 21 July 2022
NCT03776630	Exploring the Potential of Novel Biomarkers Based on Plasma microRNAs for a Better Management of Pelvic Gynecologic Tumors France	Ovarian cancer Endometrial cancer Blood sample	Interventional. Allocation: Non-Randomized Intervention Model: Parallel Assignment Masking: None (open label) Primary purpose: Diagnostic Active, not recruiting	Patients’ diagnosis with ovarian cancer and endometrial cancer	It will aim to investigate the links of the 5-miR index with classical predictors of lymph node involvement in the context of endometrial cancer. It also will aim to validate multiplexed homogenous miR detection based on RCA-FRET compared to conventional RT-qPCR in plasma samples.	24 September 2018 30 June 2022
NCT01119573	MicroRNAs associated with lymph node metastasis in endometrial cancer. United State	Endometrial cancer. Tumor tissue samples from endometrial cancer patients.	Observational. Unknown status	Not provided	Association between microRNA expression and lymph node metastasis	6 May 2010 22 June 2010
NCT04010487	A multi-omics study on the pathogenesis of malignant transformation of adenomyosis. China	Adenomyosis, endometrial cancer, ectopic endometrial tissue, eutopic endometrium	Observational, case-control, retrospective. Recruiting	Group 1. Patients pathologically conformed endometrial carcinoma arising in adenomyosis. Group 2. Patients pathologically diagnosed with adenomyosis	Frequencies of somatic driving mutations. Frequencies of alteration of RNA expression.	3 July 2019 18 July 2019
NCT02983279	Caloric restriction for oncology research: pre-operative caloric restriction prior to definitive oncologic surgery. United State	Breast carcinoma, endometrial carcinoma, prostate carcinosarcoma	Interventional, Single group assignment, Completed	Patients diagnosed with prostate, endometrial or breast cancer.	Change in miR-21 expression assessed in serum will be evaluated by a two-sided paired t-test.	14 November 2016 6 January 2023
NCT05292573	MicroRNAs as biomarkers of predicting future endometrial malignancy and longitudinal follow-up with randomized intervention in women with endometrial hyperplasia without atypia. Taiwan	Endometrial tissues and sera	Interventional, phase 3. Randomized, parallel assignment. Recruiting.	This study will prospectively enroll a total of 1000 patients with simple hyperplasia/complex hyperplasia without atypia to endometrial cancer of the 1989–2011 cohort	To evaluate the ROC (operating characteristic curve) of the prediction microRNA panel of 3 microRNAs.	8 April 2021 23 March 2022
NCT03742869	Multi-omics Study on the Human Papillomavirus Integration and Tumorigenesis of Uterine Cervical Adenocarcinoma. China	Samples with DNA from peripheral venous blood and 50 μL cancer tissue and tissue adjacent to cancer will be collected from eligible patients.	Observational Prospective case-control Unknown status	All patients confirmed primary adenocarcinoma of the uterine cervix	Alteration of mRNA, miRNA, and lncRNA pattern expression in patients with and without HPV integration	11 November 2018 23 November 2021

Clinical trial information obtained from: http://clinicaltrials.gov/ct2/home (14 April 2021 accessed).

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
