# Peer review of "Regulatory Genetic Networks by microRNAs: Exploring Genomic Signatures in Cervical Cancer"

_biomedicines, 2025, doi:10.3390/biomedicines13061457_

Round 1
Reviewer 1 Report
Comments and Suggestions for Authors
This manuscript represents a comprehensive review of the role of miRNAs in cervical cancer. The article includes an extensive collection of data. The topic is current and relevant for research, with good documentation. But it is a difficult article to navigate, being an extensive compilation of data. Is there any real clinical application of these miRNAs? Some schemes of miRNA interactions would make the article easier to navigate. I do not think it is relevant here to mention so many details for each study. Rather, the conclusions of each article would be more useful for the reader. It would be more useful to mention some inclusion and exclusion criteria for the compiled articles. Not being a geneticist by profession, I make these comments, this article being extremely useful for gynecologists, oncologists, laboratories, not only for geneticists
Author Response
REVIEWER # 3
Point 1. The reviewer said:
This manuscript represents a comprehensive review of the role of miRNAs in cervical cancer. The article includes an extensive collection of data. The topic is current and relevant for research, with good documentation. But it is a difficult article to navigate, being an extensive compilation of data. Is there any real clinical application of these miRNAs? Some schemes of miRNA interactions would make the article easier to navigate. I do not think it is relevant here to mention so many details for each study. Rather, the conclusions of each article would be more useful for the reader. It would be more useful to mention some inclusion and exclusion criteria for the compiled articles. Not being a geneticist by profession, I make these comments, this article being extremely useful for gynecologists, oncologists, laboratories, not only for geneticists
Response to point 1. Thank you very much for these comments.
The clinical application of these miRNAs is contained in “5. Gene therapy clinical trials with microRNAs for uterine cancers” section.
We included the datails for each study to ensure transparency and reproducibility.
A Prima-style flowchart was improved as figure 1 to mention some inclusion and exclusion criteria for the compiled articles.
In additional, in the introduction section, we included the next sentence to clarify the description of the methodology used to select the studies included in the review article.
“Several previous studies have identified microRNAs as potential targets for different therapeutic strategies, associated with HPV infection, or have linked specific microRNAs to cervical carcinogenesis process. By perming Medline, we search for the keywords cervical cancer and microRNA signature (April 14, 2021). In this work, only the studies that indicate the platforms used for microRNA analysis, several experimental conditions, regulatory genetic networks modulated by microRNAs, microRNA-mRNA regulatory modules, and clinical trials with microRNAs for uterine cancers were included. We analyzed from these publications the microRNAs those that were reported to be differentially regulated in cervical cancer tissue or in in vitro models. This is a critical issue since HPV E6/E7 expression levels are tightly controlled in HPV-positive cancer cells and it is not clear how this relates to the E6/E7 levels obtained. Thus, we analyzed the experimental approaches that have addressed the question whether the actual cellular microRNA composition and regulatory genetic networks of HPV-positive cancer cells depends on endogenous E6/E7 expression. Thus, it is crucial to investigate the cellular regulatory genetic networks by microRNAs, within the cellular background of HPV-transformed cervical cancer cells.
Although several studies have provided significant findings about microRNA signatures and cervical cancer, conclusions about the central molecular mechanisms behind the disease were not reached because this type of information requires an integrative multi-omics approach. The computational integration of biomolecular networks with data from different omic levels represents the core of research in the field of regulatory genetic networks. This interdisciplinary field provides valuable information on genome reprogramming under disease conditions and relevant biological entities that might be considered potential diagnostic or therapeutic targets for cervical cancer. Furthermore, it is important to investigate target hubs subject to coordinated regulation by dozens of microRNAs, as these are often key regulators of biological processes. Moreover, increasing evidence has indicated that microRNA-mediated gene regulatory networks are critical for inferring cooperative microRNA regulation in cancer. The complexity of such networks results from the multiplicity of microRNA-target interactions, which often engage in reciprocal feedback and feedforward loops.
In this review, 70 different studies have been reported with different microRNA signatures, 15 databases, and 9 clinical trials were included in this study. The microRNA signatures experimental reports were analyzed which were proposed by several research groups to be associated with cervical cancer. Figure 1 shows a Prisma flow diagram with detailed description of the methodology used to select the studies included in this review article.”
Reviewer 2 Report
Comments and Suggestions for Authors
The review manuscript entitled “Regulatory Genetic Networks by microRNAs: Exploring Genomic Signatures in Cervical Cancer” provides a comprehensive overview of the current state of research on microRNAs in cervical cancer.
The authors have reviewed a substantial amount of information, highlighting the potential of microRNAs as diagnostic and prognostic biomarkers, as well as their involvement in the complex regulatory networks associated with cervical carcinogenesis.
For a better understanding and text organization I have some suggestions:
- The authors do not present a detailed description of the methodology used to select the studies included in the review. It is important to describe what kind of review they have proposed as well as the criteria of articles selection. A PRISMA-style flowchart could improve clarity regarding the criteria utilized.
- Section 2 – should be subdivided in subheadings, according to the biological sample analyzed, as it is organized in table 1 (cervical tissues and circulating).
- Table 1 – Table 1 could be organized by biospecimen, first all studies concerning cervical tissues and after circulating ones, this would make it easier to compare and interpret the findings.
- Section 3 – In this section authors describe many networks that are modulated by miRNAs. In order to provide a better understanding and a figure stating the main molecules and their respective pathways could improve the section comprehension.
Author Response
Cuernavaca, Morelos, México, June 03, 2025
Ms. Amity Liu
Assistant Editor
Biomedicines
Dear Ms. Amity Liu
Thank you very much for giving me the opportunity to submit our Manuscript ID: biomedicines-3525002 Type of manuscript: Review Title: Regulatory Genetic Networks by microRNAs: Exploring Genomic Signatures in Cervical Cancer, to the special issue "Advanced Research in Gynecologic Oncology", to be published in the "Biomedicines" (ISSN 2227-9059, IF 3.9).
We are very interested in re-submitting our manuscript and I am attaching a cover letter to explain, point by point, to the comments raised by the reviewers. In addition, I ensured that all changes in the manuscript are indicated in the text in bold and green by highlighting or using track changes.
Thank you very much for your kind attention
I look forward to receiving your positive response in this regard.
Best regards
Oscar Peralta-Zaragoza. Ph. D.
Corresponding Author
Direction of Chronic Infections and Cancer,
Research Center in Infection Diseases,
National Institute of Public Health.
Av. Universidad No. 655, Cerrada los Pinos
y Caminera. Colonia Santa María Ahuacatitlán,
Cuernavaca, Morelos, México 62100.
Tel: (+52)-777-3293000 ext. 2406
E-mail: operalta@insp.mx
Please find enclosed the cover letter to explain, point by point, to the comments raised by the reviewers.
REVIEWER # 1
Point 1. The reviewer said:
The review manuscript entitled “Regulatory Genetic Networks by microRNAs: Exploring Genomic Signatures in Cervical Cancer” provides a comprehensive overview of the current state of research on microRNAs in cervical cancer.
The authors have reviewed a substantial amount of information, highlighting the potential of microRNAs as diagnostic and prognostic biomarkers, as well as their involvement in the complex regulatory networks associated with cervical carcinogenesis.
For a better understanding and text organization I have some suggestions:
The authors do not present a detailed description of the methodology used to select the studies included in the review. It is important to describe what kind of review they have proposed as well as the criteria of articles selection. A PRISMA-style flowchart could improve clarity regarding the criteria utilized.
Response to point 1. Thank you very much for these comments.
A Prima-style flowchart was improved as figure 1 to mention some inclusion and exclusion criteria for the compiled articles. Figure 1.
In additional, In the introduction section, we included the next sentence to clarify the description of the methodology used to select the studies included in the review article.
“Several previous studies have identified microRNAs as potential targets for different therapeutic strategies, associated with HPV infection, or have linked specific microRNAs to cervical carcinogenesis process. By perming Medline, we search for the keywords cervical cancer and microRNA signature (April 14, 2021). In this work, only the studies that indicate the platforms used for microRNA analysis, several experimental conditions, regulatory genetic networks modulated by microRNAs, microRNA-mRNA regulatory modules, and clinical trials with microRNAs for uterine cancers were included. We analyzed from these publications the microRNAs those that were reported to be differentially regulated in cervical cancer tissue or in in vitro models. This is a critical issue since HPV E6/E7 expression levels are tightly controlled in HPV-positive cancer cells and it is not clear how this relates to the E6/E7 levels obtained. Thus, we analyzed the experimental approaches that have addressed the question whether the actual cellular microRNA composition and regulatory genetic networks of HPV-positive cancer cells depends on endogenous E6/E7 expression. Thus, it is crucial to investigate the cellular regulatory genetic networks by microRNAs, within the cellular background of HPV-transformed cervical cancer cells.
Although several studies have provided significant findings about microRNA signatures and cervical cancer, conclusions about the central molecular mechanisms behind the disease were not reached because this type of information requires an integrative multi-omics approach. The computational integration of biomolecular networks with data from different omic levels represents the core of research in the field of regulatory genetic networks. This interdisciplinary field provides valuable information on genome reprogramming under disease conditions and relevant biological entities that might be considered potential diagnostic or therapeutic targets for cervical cancer. Furthermore, it is important to investigate target hubs subject to coordinated regulation by dozens of microRNAs, as these are often key regulators of biological processes. Moreover, increasing evidence has indicated that microRNA-mediated gene regulatory networks are critical for inferring cooperative microRNA regulation in cancer. The complexity of such networks results from the multiplicity of microRNA-target interactions, which often engage in reciprocal feedback and feedforward loops.
In this review, 70 different studies have been reported with different microRNA signatures, 15 databases, and 9 clinical trials were included in this study. The microRNA signatures experimental reports were analyzed which were proposed by several research groups to be associated with cervical cancer. Figure 1 shows a Prisma flow diagram with detailed description of the methodology used to select the studies included in this review article.”
Point 2. The reviewer said:
Section 2 – should be subdivided in subheadings, according to the biological sample analyzed, as it is organized in table 1 (cervical tissues and circulating).
Response to point 2. Thank you very much for these comments.
We subdivided section 2 into: 2.1. Genomic signature of microRNAs in cervical tissues, and 2.2. Genomic signature of microRNAs circulating.
Point 3. The reviewer said:
Table 1 – Table 1 could be organized by biospecimen, first all studies concerning cervical tissues and after circulating ones, this would make it easier to compare and interpret the findings.
Response to point 3. Thank you very much for these comments.
Table I was organized by biospecimen, first all studies concerning microRNAs in cervical tissues and after microRNAs circulating.
Point 4. The reviewer said:
Section 3 – In this section authors describe many networks that are modulated by miRNAs. In order to provide a better understanding and a figure stating the main molecules and their respective pathways could improve the section comprehension.
Response to point 4. Thank you very much for these comments.
In section 3, we did a review, and we improved section comprehension more.
REVIEWER # 2
Point 1. The reviewer said:
The article "Regulatory Genetic Networks by microRNAs: Exploring Genomic Signatures in Cervical Cancer " provides valuable information in the field; however, several issues must be addressed to meet the standards for publication:
- The authenticate analysis indicates that the similarity score of 57% is unusually high for a scientific manuscript.
Response to point 1. Thank you very much for these comments.
This version of manuscript contains an updated version with reducing the similarity index according to publication requirements of Biomedicines Journal.
Point 2. The reviewer said:
- Although the manuscript is structured as a review article, it references bioinformatic databases, predictive analyses, and a significant integration of data from the scientific literature. Despite this, there is no dedicated “Materials and Methods” section to ensure transparency and reproducibility.
Response to point 2. Thank you very much for these comments.
We did not dedicate a “Material and Methods” section because our work effectively is a review article. Furthermore, in Instructions for Authors from Biomedicines Journal does not request this section for review articles.
“Review: Reviews offer a comprehensive analysis of the existing literature within a field of study, identifying current gaps or problems. They should be critical and constructive and provide recommendations for future research. No new, unpublished data should be presented. The structure can include an Abstract, Keywords, Introduction, Relevant Sections, Discussion, Conclusions, and Future Directions.”
Nevertheless, to ensure transparency and reproducibility of this review article, in the introduction section we included the next sentence:
“Several previous studies have identified microRNAs as potential targets for different therapeutic strategies, associated with HPV infection, or have linked specific microRNAs to cervical carcinogenesis process. By perming Medline, we search for the keywords cervical cancer and microRNA signature (April 14, 2021). In this work, only the studies that indicate the platforms used for microRNA analysis, several experimental conditions, regulatory genetic networks modulated by microRNAs, microRNA-mRNA regulatory modules, and clinical trials with microRNAs for uterine cancers were included. We analyzed from these publications the microRNAs those that were reported to be differentially regulated in cervical cancer tissue or in in vitro models. This is a critical issue since HPV E6/E7 expression levels are tightly controlled in HPV-positive cancer cells and it is not clear how this relates to the E6/E7 levels obtained. Thus, we analyzed the experimental approaches that have addressed the question whether the actual cellular microRNA composition and regulatory genetic networks of HPV-positive cancer cells depends on endogenous E6/E7 expression. Thus, it is crucial to investigate the cellular regulatory genetic networks by microRNAs, within the cellular background of HPV-transformed cervical cancer cells.
Although several studies have provided significant findings about microRNA signatures and cervical cancer, conclusions about the central molecular mechanisms behind the disease were not reached because this type of information requires an integrative multi-omics approach. The computational integration of biomolecular networks with data from different omic levels represents the core of research in the field of regulatory genetic networks. This interdisciplinary field provides valuable information on genome reprogramming under disease conditions and relevant biological entities that might be considered potential diagnostic or therapeutic targets for cervical cancer. Furthermore, it is important to investigate target hubs subject to coordinated regulation by dozens of microRNAs, as these are often key regulators of biological processes. Moreover, increasing evidence has indicated that microRNA-mediated gene regulatory networks are critical for inferring cooperative microRNA regulation in cancer. The complexity of such networks results from the multiplicity of microRNA-target interactions, which often engage in reciprocal feedback and feedforward loops.
In this review, 70 different studies have been reported with different microRNA signatures, 15 databases, and 9 clinical trials were included in this study. The microRNA signatures experimental reports were analyzed which were proposed by several research groups to be associated with cervical cancer. Figure 1 shows a Prisma flow diagram with detailed description of the methodology used to select the studies included in this review article.”
Point 3. The reviewer said:
- The manuscript is dense, informative, and well-referenced; however, it largely compiles information from previous studies without the authors' clearly defined original contribution. It is recommended that a dedicated section for critical discussion be included.
Response to point 3. Thank you very much for these comments.
In the discussion section, we included the next sentence:
“In addition, we did an analysis to identify which microRNAs can serve as a genomic signature for patients with cervical cancer. The novelty and interesting thing about this analysis is that it is possible to identify circulating microRNAs, that is, in patients' bio-fluids. Therefore, this proposal represents an advantage for patients because it is a non-invasive intervention, since the expression profile of microRNAs in different patient biofluids can be identified. These microRNA genomic signatures can be validated by the different integrative multi-omics approach strategies.”.
REVIEWER # 3
Point 1. The reviewer said:
This manuscript represents a comprehensive review of the role of miRNAs in cervical cancer. The article includes an extensive collection of data. The topic is current and relevant for research, with good documentation. But it is a difficult article to navigate, being an extensive compilation of data. Is there any real clinical application of these miRNAs? Some schemes of miRNA interactions would make the article easier to navigate. I do not think it is relevant here to mention so many details for each study. Rather, the conclusions of each article would be more useful for the reader. It would be more useful to mention some inclusion and exclusion criteria for the compiled articles. Not being a geneticist by profession, I make these comments, this article being extremely useful for gynecologists, oncologists, laboratories, not only for geneticists
Response to point 1. Thank you very much for these comments.
The clinical application of these miRNAs is contained in “5. Gene therapy clinical trials with microRNAs for uterine cancers” section.
We included the details for each study to ensure transparency and reproducibility.
A Prima-style flowchart was improved as figure 1 to mention some inclusion and exclusion criteria for the compiled articles.
In additional, in the introduction section, we included the next sentence to clarify the description of the methodology used to select the studies included in the review article.
“Several previous studies have identified microRNAs as potential targets for different therapeutic strategies, associated with HPV infection, or have linked specific microRNAs to cervical carcinogenesis process. By perming Medline, we search for the keywords cervical cancer and microRNA signature (April 14, 2021). In this work, only the studies that indicate the platforms used for microRNA analysis, several experimental conditions, regulatory genetic networks modulated by microRNAs, microRNA-mRNA regulatory modules, and clinical trials with microRNAs for uterine cancers were included. We analyzed from these publications the microRNAs those that were reported to be differentially regulated in cervical cancer tissue or in in vitro models. This is a critical issue since HPV E6/E7 expression levels are tightly controlled in HPV-positive cancer cells and it is not clear how this relates to the E6/E7 levels obtained. Thus, we analyzed the experimental approaches that have addressed the question whether the actual cellular microRNA composition and regulatory genetic networks of HPV-positive cancer cells depends on endogenous E6/E7 expression. Thus, it is crucial to investigate the cellular regulatory genetic networks by microRNAs, within the cellular background of HPV-transformed cervical cancer cells.
Although several studies have provided significant findings about microRNA signatures and cervical cancer, conclusions about the central molecular mechanisms behind the disease were not reached because this type of information requires an integrative multi-omics approach. The computational integration of biomolecular networks with data from different omic levels represents the core of research in the field of regulatory genetic networks. This interdisciplinary field provides valuable information on genome reprogramming under disease conditions and relevant biological entities that might be considered potential diagnostic or therapeutic targets for cervical cancer. Furthermore, it is important to investigate target hubs subject to coordinated regulation by dozens of microRNAs, as these are often key regulators of biological processes. Moreover, increasing evidence has indicated that microRNA-mediated gene regulatory networks are critical for inferring cooperative microRNA regulation in cancer. The complexity of such networks results from the multiplicity of microRNA-target interactions, which often engage in reciprocal feedback and feedforward loops.
In this review, 70 different studies have been reported with different microRNA signatures, 15 databases, and 9 clinical trials were included in this study. The microRNA signatures experimental reports were analyzed which were proposed by several research groups to be associated with cervical cancer. Figure 1 shows a Prisma flow diagram with detailed description of the methodology used to select the studies included in this review article.”
All the comments suggested by the Reviewers were included and are indicated in bold and green in the manuscript. This version of the manuscript includes a revision of the English language edition.
Thank you very much for your kind attention
I look forward to receiving your positive response in this regard.
Best regards
Oscar Peralta-Zaragoza. Ph. D.
Corresponding Author
Direction of Chronic Infections and Cancer,
Research Center in Infection Diseases,
National Institute of Public Health.
Av. Universidad No. 655, Cerrada los Pinos
y Caminera. Colonia Santa María Ahuacatitlán,
Cuernavaca, Morelos, México 62100.
Tel: (+52)-777-3293000 ext. 2406
E-mail: operalta@insp.mx

Reviewer 3 Report
Comments and Suggestions for Authors
The article "Regulatory Genetic Networks by microRNAs: Exploring Genomic Signatures in Cervical Cancer " provides valuable information in the field; however, several issues must be addressed to meet the standards for publication:
1. The ithenticate analysis indicates that the similarity score of 57% is unusually high for a scientific manuscript.
2. Although the manuscript is structured as a review article, it references bioinformatic databases, predictive analyses, and a significant integration of data from the scientific literature. Despite this, there is no dedicated “Materials and Methods” section to ensure transparency and reproducibility.
3. The manuscript is dense, informative, and well-referenced; however, it largely compiles information from previous studies without the authors' clearly defined original contribution. It is recommended that a dedicated section for critical discussion be included.
The language is generally appropriate, but some sentences are overly complex or repetitive. A thorough linguistic revision is advised.
Author Response
sb
REVIEWER # 2
Point 1. The reviewer said:
The article "Regulatory Genetic Networks by microRNAs: Exploring Genomic Signatures in Cervical Cancer " provides valuable information in the field; however, several issues must be addressed to meet the standards for publication:
- The authenticate analysis indicates that the similarity score of 57% is unusually high for a scientific manuscript.
Response to point 1. Thank you very much for these comments.
This version of manuscript contains an updated version with reducing the similarity index according to publication requirements of Biomedicines Journal.
Point 2. The reviewer said:
- Although the manuscript is structured as a review article, it references bioinformatic databases, predictive analyses, and a significant integration of data from the scientific literature. Despite this, there is no dedicated “Materials and Methods” section to ensure transparency and reproducibility.
Response to point 2. Thank you very much for these comments.
We did not dedicate a “Material and Methods” section because our work effectively is a review article. Furthermore, in Instructions for Authors from Biomedicines Journal does not request this section for review articles.
“Review: Reviews offer a comprehensive analysis of the existing literature within a field of study, identifying current gaps or problems. They should be critical and constructive and provide recommendations for future research. No new, unpublished data should be presented. The structure can include an Abstract, Keywords, Introduction, Relevant Sections, Discussion, Conclusions, and Future Directions.”
Nevertheless, to ensure transparency and reproducibility of this review article, in the introduction section we included the next sentence:
“Several previous studies have identified microRNAs as potential targets for different therapeutic strategies, associated with HPV infection, or have linked specific microRNAs to cervical carcinogenesis process. By perming Medline, we search for the keywords cervical cancer and microRNA signature (April 14, 2021). In this work, only the studies that indicate the platforms used for microRNA analysis, several experimental conditions, regulatory genetic networks modulated by microRNAs, microRNA-mRNA regulatory modules, and clinical trials with microRNAs for uterine cancers were included. We analyzed from these publications the microRNAs those that were reported to be differentially regulated in cervical cancer tissue or in in vitro models. This is a critical issue since HPV E6/E7 expression levels are tightly controlled in HPV-positive cancer cells and it is not clear how this relates to the E6/E7 levels obtained. Thus, we analyzed the experimental approaches that have addressed the question whether the actual cellular microRNA composition and regulatory genetic networks of HPV-positive cancer cells depends on endogenous E6/E7 expression. Thus, it is crucial to investigate the cellular regulatory genetic networks by microRNAs, within the cellular background of HPV-transformed cervical cancer cells.
Although several studies have provided significant findings about microRNA signatures and cervical cancer, conclusions about the central molecular mechanisms behind the disease were not reached because this type of information requires an integrative multi-omics approach. The computational integration of biomolecular networks with data from different omic levels represents the core of research in the field of regulatory genetic networks. This interdisciplinary field provides valuable information on genome reprogramming under disease conditions and relevant biological entities that might be considered potential diagnostic or therapeutic targets for cervical cancer. Furthermore, it is important to investigate target hubs subject to coordinated regulation by dozens of microRNAs, as these are often key regulators of biological processes. Moreover, increasing evidence has indicated that microRNA-mediated gene regulatory networks are critical for inferring cooperative microRNA regulation in cancer. The complexity of such networks results from the multiplicity of microRNA-target interactions, which often engage in reciprocal feedback and feedforward loops.
In this review, 70 different studies have been reported with different microRNA signatures, 15 databases, and 9 clinical trials were included in this study. The microRNA signatures experimental reports were analyzed which were proposed by several research groups to be associated with cervical cancer. Figure 1 shows a Prisma flow diagram with detailed description of the methodology used to select the studies included in this review article.”
Point 3. The reviewer said:
- The manuscript is dense, informative, and well-referenced; however, it largely compiles information from previous studies without the authors' clearly defined original contribution. It is recommended that a dedicated section for critical discussion be included.
Response to point 3. Thank you very much for these comments.
In the discussion section, we included the next sentence:
“In addition, we did an analysis to identify which microRNAs can serve as a genomic signature for patients with cervical cancer. The novelty and interesting thing about this analysis is that it is possible to identify circulating microRNAs, that is, in patients' biofluids. Therefore, this proposal represents an advantage for patients because it is a non-invasive intervention, since the expression profile of microRNAs in different patient biofluids can be identified. These microRNA genomic signatures can be validated by the different integrative multi-omics approach strategies”.
Round 2
Reviewer 3 Report
Comments and Suggestions for Authors
Following the review of the revised manuscript, I note that the similarity index remains unchanged compared to the previous version. Although the authors have submitted a new version of the article, no substantial changes have been made to reduce the degree of textual overlap with already published sources.
The persistence of a high similarity index raises serious concerns regarding the originality of the content, compliance with academic best practices, and adherence to the scientific integrity standards required by your journal. For this reason, I believe the manuscript cannot be accepted for publication in its current form.
I recommend that the authors thoroughly revise the text, rephrase the problematic sections, and ensure the manuscript reflects an original contribution that can be considered for publication.
Author Response

(The authors gave the same response as above.)

Round 3
Reviewer 3 Report
Comments and Suggestions for Authors
The article demonstrates scientific value and coherence; however, given the current similarity index of 28%, it is recommended to reduce this percentage to a level that meets the originality standards required by reputable scientific journals, enabling publication.